# Small-molecule suppression of calpastatin degradation reduces neuropathology in models of Huntington's disease

Di Hu 📧 [1], Xiaoyan Sun[1], Anniefer Magpusao[2], Yuriy Fedorov[2], Matthew Thompson[2], Benlian Wang[3], Kathleen Lundberg[3], Drew J. Adams[2✉] & Xin Qi 📧 [1✉]

Mitochondrial dysfunction is a common hallmark of neurological disorders, and reducing mitochondrial damage is considered a promising neuroprotective therapeutic strategy. Here, we used high-throughput small molecule screening to identify CHIR99021 as a potent enhancer of mitochondrial function. CHIR99021 improved mitochondrial phenotypes and enhanced cell viability in several models of Huntington's disease (HD), a fatal inherited neurodegenerative disorder. Notably, CHIR99201 treatment reduced HD-associated neuropathology and behavioral defects in HD mice and improved mitochondrial function and cell survival in HD patient-derived neurons. Independent of its known inhibitory activity against glycogen synthase kinase 3 (GSK3), CHIR99021 treatment in HD models suppressed the proteasomal degradation of calpastatin (CAST), and subsequently inhibited calpain activation, a well-established effector of neural death, and Drp1, a driver of mitochondrial fragmentation. Our results established CAST-Drp1 as a druggable signaling axis in HD pathogenesis and highlighted CHIR99021 as a mitochondrial function enhancer and a potential lead for developing HD therapies.

[1] Department of Physiology & Biophysics, Case Western Reserve University School of Medicine, Cleveland, OH, USA. [2] Department of Genetics, Case Western Reserve University School of Medicine, Cleveland, OH, USA. [3] Proteomics Center, Case Western Reserve University School of Medicine, Cleveland, OH, USA. ✉email: dja59@case.edu; xxq38@case.edu

Normal mitochondrial function is essential for energy production, lipid metabolism, calcium buffering, and redox regulation, which govern cell growth, proliferation, and survival[1]. Because neurons have high energetic demands and limited regenerative capacity, mitochondrial integrity disruption adversely affects neuronal function. In particular, dysregulation of mitochondrial fusion and fission, biogenesis failure, or inefficient oxidative phosphorylation may lead to shortened dendritic trees and axons, the downregulation of neurotransmitter turnover, inflammatory responses, and ultimately neuronal loss[1]. Thus, mitochondrial dysfunction is hypothesized to play pathogenic roles and promote neurodegeneration in Alzheimer's disease (AD), Parkinson's disease (PD), Huntington's disease (HD), and amyotrophic lateral sclerosis (ALS)[2]. Approaches improving mitochondrial function may be effective therapeutic strategies for treating such diseases; however, limited efforts have been made to identify mitochondrial enhancers that manipulate pathological processes during neurodegeneration.

HD is a fatal neurological disorder caused by a trinucleotide CAG repeat expansion in the N-terminal exon 1 of the huntingtin (*Htt*) gene. HD manifests as uncontrolled involuntary movements (chorea and dyskinesia) accompanied by progressive motor and cognitive deficits, psychiatric disturbance, and dementia[3]. Pathologically, HD is associated with a diffuse loss of neurons, particularly those at the striatum and cortex. Medium spiny neurons (MSNs) in the striatum containing γ-aminobutyric acid and enkephalin are affected early in the disease and are the primary neurons targeted in HD[4]. Although the cause of HD has been identified and genetic tests are available to identify individuals carrying the mutation who will ultimately succumb to the disease, currently no therapies exist to slow or prevent disease progression; only symptomatic treatments are available with limited impact[5].

Most evidence indicates that mitochondrial dysfunction is an early prominent feature in susceptible neurons in the brains of patients with HD and plays a critical role in its pathogenesis[6]. HD experimental models in cell culture and animals exhibit mitochondrial dysfunction, including mitochondrial depolarization, oxidative stress, bioenergetic defects, and mitochondrial DNA depletion[7]. Decreased mitochondrial oxygen consumption, depleted ATP, reduced glucose metabolism, and elevated lactate concentrations are all observed in HD patients and animals before disease onset[7]. Moreover, mutant Htt protein (mtHtt) impairs mitochondrial integrity via multiple pathways; e.g., it interferes with peroxisome proliferator-activated receptor-γ (PPARγ) coactivator 1-α (PGC1α)-mediated transcription of mitochondrial bioenergetic-related genes[8]. Toxic mtHtt is also reported to accumulate on the mitochondrial membrane to disrupt the permeability transition pores and oxidative phosphorylation, leading to mitochondrial uncoupling and bioenergetic deficits[9]. In addition, mtHtt activates the mitochondrial fission protein dynamin-related protein 1 (Drp1), which causes excessive mitochondrial fragmentation and subsequent neuronal death[10,11].

Pharmacological reagents that lessen mitochondrial dysfunction provide neuroprotection in various HD models. In previous work, we rationally designed cell-penetrating peptides that inhibited mitochondrial damage. Both P110, a peptide inhibitor that selectively suppressed Drp1 hyperactivation[12], and HV-3, which specifically blocked mtHtt/valosin-containing protein interaction[13], rescued mitochondrial damage, reduced neuronal cell death, and attenuated behavioral deficits in HD transgenic mice[13,14]. Other approaches improving mitochondrial bioenergetics by targeting PGC1α/PPARγ mitigated the progression of HD-related symptoms[15]. These findings not only provided evidence of a causal role for mitochondrial damage in HD pathogenesis but also strongly supported the notion that reversing mitochondrial dysfunction can reduce neuronal pathology and potentially provide therapeutic benefits for patients with HD.

In this study, we used a cell-based model of HD to screen small molecules for their ability to suppress mitochondrial damage and cell death and identified CHIR99021 as a potent enhancer of mitochondrial activity. Subsequent validation assays demonstrated that CHIR99021 treatment rescued mitochondrial membrane potential (MMP), bioenergetics, and mitochondrial morphology and improved cell viability in HD mouse- and patient-derived neurons. Notably, long-term administration of CHIR99021 reduced the progression of neuropathology and the development of motor deficits in both R6/2 and YAC128 HD mouse models. Mechanistically, we demonstrated that the protective efficacy of CHIR99021 was independent of its glycogen synthase kinase 3 (GSK3) inhibitory activity but was dependent on calpastatin (CAST), a specific endogenous inhibitor of the cysteine protease, calpain I. Treatment with CHIR99201 prevented CAST protein degradation via the ubiquitin–proteasome system (UPS), restored CAST protein expression, and subsequently suppressed calpain activity. Finally, we established that CAST stabilization by CHIR99021 restrained Drp1 mitochondrial recruitment and reduced mitochondrial fragmentation, leading to enhanced mitochondrial function and reduced HD-associated neuropathology. Therefore, our findings reveal a druggable CAST–Drp1 signaling axis in the pathogenesis of HD and suggest that a CHIR99021-like small molecule might be a useful lead in the development of HD therapy.

## Results

**A robust high-throughput screening (HTS) platform identified CHIR99021 as an enhancer of mitochondrial function in HD cells.** Using mouse striatal cells (HdhQ111) derived from HD knock-in transgenic mice[16], we developed a 384-well format assay to measure MMP, which is the driving force for ATP synthesis at the last step of oxidative phosphorylation[17]. Since this process is compromised early during stress responses, MMP is a faithful indicator of mitochondrial health and cell toxicity[17]. To detect MMP, we used tetramethylrhodamine (TMRM), a fluorescence reporter that accumulates in mitochondria in proportion to MMP. HdhQ111 striatal cells showed significantly reduced TMRM fluorescence relative to isogenic wild-type (WT) HdhQ7 cells (Supplementary Fig. 1a), consistent with mitochondrial depolarization in this model[13,14]. We then performed HTS using a 53,000 small-molecule library and identified 54 hits with half maximal effective concentration (EC$_{50}$) values ranging from 0.3 to 5 μM (Supplementary Fig. 1a, b). To prioritize hit molecules rendering HD cells more resistant to cytotoxic stresses, we measured viability in HdhQ111 cells 16 h after serum withdrawal. HdhQ111 cells showed elevated cell death rates relative to HdhQ7 cells under the same conditions (Supplementary Fig. 1a), consistent with previous studies[14,18]. Although several molecules exerted positive effects in both assays, the GSK3 inhibitor, CHIR99021, exhibited the clearest protective effects on both MMP and cell survival (Supplementary Fig. 1). Notably, CHIR99021 treatment in HdhQ111 cells improved MMP to levels observed in WT HdhQ7 cells and suppressed cell death following serum withdrawal, with an EC$_{50}$ of 3–3.5 μM (Fig. 1a and Supplementary Fig. 1c). CHIR99021 was therefore selected for follow-up studies in HD models.

**CHIR99021 restored mitochondrial function in HD mouse- and patient-derived neuronal cells.** We validated the protective effects of CHIR99021 using a series of assays in HD neuronal cells. Treatment with CHIR99021 at 3 μM reduced mitochondrial superoxide production (Fig. 1b) and enhanced the cell survival of

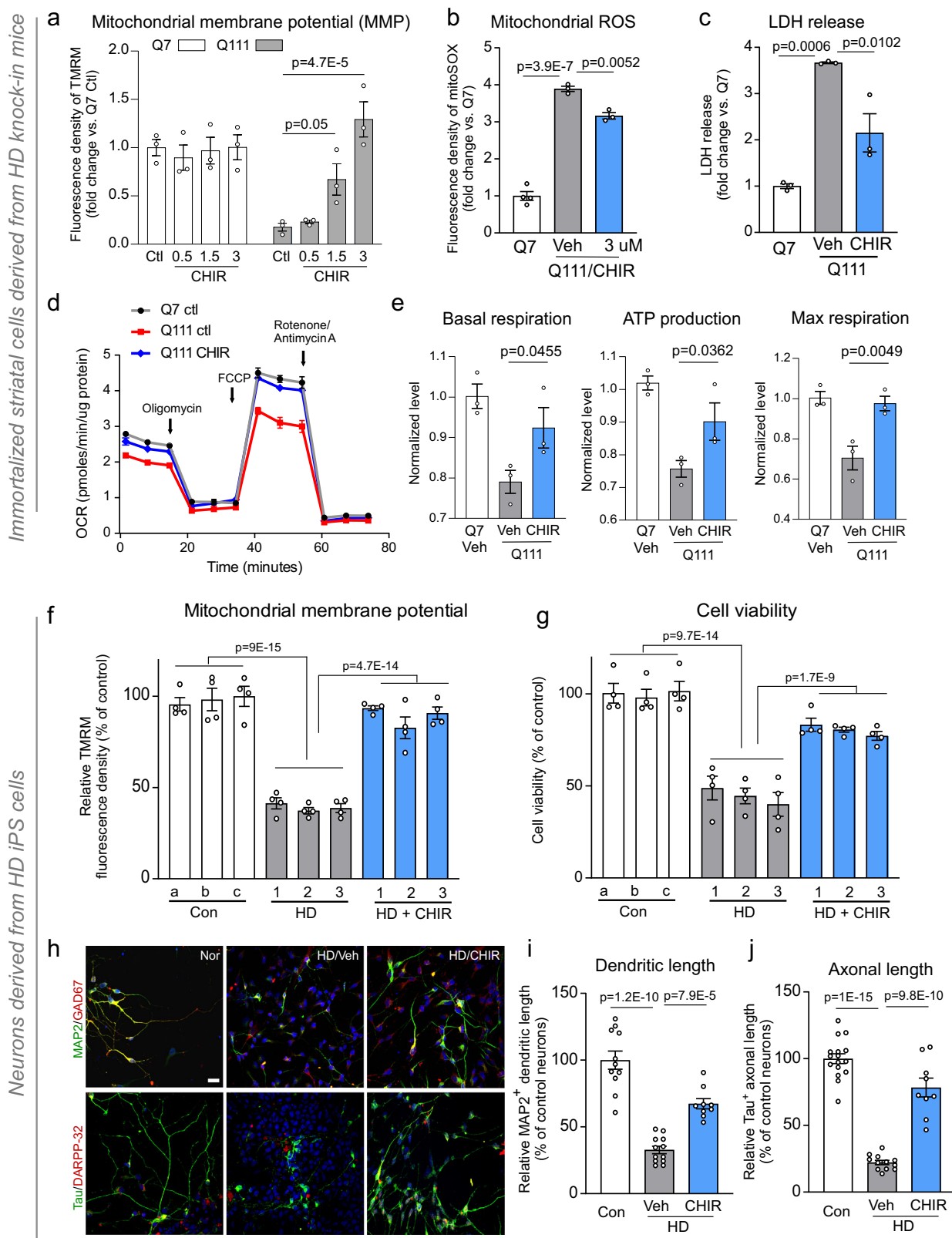

serum-starved HdhQ111 striatal cells using an orthogonal read-out of cell viability: lactate dehydrogenase (LDH) release (Fig. 1c). Furthermore, CHIR99021 treatment enhanced the oxygen consumption rate in HdhQ111 cells relative to cells treated with dimethylsulfoxide (DMSO) as vehicle (Fig. 1d), as evidenced by improved basal respiration, maximal respiration, and ATP production (Fig. 1e). In contrast, CHIR99021 had no effects on

mitochondrial respiration in HdhQ7 cells (Supplementary Fig. 1d).

HD is defined by the selective loss of striatal GABAergic MSNs[19]. We previously differentiated HD patient-induced pluripotent stem (iPS) cells into GABAergic striatal and MSNs; their mtHtt had the same number of CAG repeats as cells from source HD patients. These cells exhibited short neurites and were

**Fig. 1 CHIR99021 reduces mitochondrial damage and cell death in HD mouse- and patient-derived neuronal cells.** HdhQ7 and HdhQ111 cells were treated with vehicle (Veh, DMSO) or CHIR99021 at the indicated doses for 2 days. **a** Cells were stained with tetramethylrhodamine (TMRM) fluorescence dye. TMRM relative fluorescence density indicates the extent of mitochondrial membrane potential (MMP). **b** Cells were stained with MitoSOX[TM] dye to assess mitochondrial superoxide (ROS) production. Histogram: the relative fluorescence density of MitoSOX[TM]. **c** Cells were subjected to a 16 h serum starvation with cell death measured by lactate dehydrogenase (LDH) release. **d** Mitochondrial respiratory activity was measured using a Seahorse XFP analyzer and a Mito stress kit. OCR oxygen consumption rate. **e** Basal respiration rate, maximal respiration rate, and ATP production are shown. Mixed striatal neurons were differentiated from iPS cells of patients with HD (1: GM04693, 41 CAG; 2: ND41656, 57 CAG; 3: ND36998, 60 CAG) and control subjects (a: HDF; b: NN0003888; c: nHDF). Twenty days after neuronal differentiation, cells were treated with DMSO (Veh) or CHIR99021 (1 μM) for 5 days. **f** MMP was assessed using a TMRM fluorescence probe ($n = 4$ biologically independent samples). **g** Neuronal viability was measured by MTT assay after the withdrawal of brain-derived neurotrophic factor (BDNF) for 12 h ($n = 4$ biologically independent samples). **h** Neurons were stained with anti-MAP2 (green)/anti-GAD67 (red) or anti-Tau (green)/anti-DARPP-32 (red) to indicate dendritic and axonal morphology, respectively. Control neurons: Nor, HD neurons: HD. Scale bar = 30 μm. **i** Quantification of MAP2$^+$ neuronal dendrite length (Con $n = 11$, HD Veh $n = 12$, HD CHIR $n = 9$ cells). **j** Quantification of Tau$^+$ neuronal axon length (Con $n = 16$, HD Veh $n = 12$, HD CHIR $n = 9$ cells). All values are reported as mean ± SEM. Data are representative of at least three independent experiments. Data were compared using one-way ANOVA with Tukey's post hoc test in **a–e**, **i**, **j**, and two-way ANOVA with Tukey's post hoc test in **f**, **g**. Exact $p$ values are shown in the figures.

sensitive to stressors[13,14]. Mitochondrial depolarization, mitochondrial fragmentation, and neuronal cell death were also evident in these patient neurons[13,14]. Thus, this patient-derived model represented a platform to validate small-molecule enhancers of mitochondrial function and survival in human neurons with HD phenotypes caused by an HD genotype. Neuronal cells differentiated from iPS cells from three HD patients consistently exhibited lower MMP and higher cell death rates following the withdrawal of brain-derived neurotrophic factor (BDNF) when compared with cells from control subjects (Fig. 1f, g). Notably, CHIR99021 treatment enhanced MMP and cell viability in patient-derived cells (Fig. 1f, g), recapitulating the efficacy of CHIR99021 in our HD mouse striatal cell line (Fig. 1a, c). Dose-dependent CHIR99021 protection toward MMP and cell viability was also observed in neurons differentiated from HD patient iPS cells (Supplementary Fig. 1e). Also, dendritic (MAP2$^+$) and axonal (Tau$^+$) lengths in HD MSNs were shorter than control neurons, however, CHIR99021 treatment significantly increased these dimensions in HD patient MSNs (Fig. 1h–j), further supporting a role for CHIR99021 in neuronal survival. Taken together, these results demonstrated that CHIR99021 enhanced mitochondrial function and cell survival in mouse- and patient-derived neuronal HD models.

**CHIR99021 was protective in HD animal models.** We next determined whether CHIR99021 treatment affected disease progression in HD mouse models. We first used HD R6/2 transgenic mice that express a fragment of human mtHtt protein and aggressively develop HD-associated pathologies, including mtHtt accumulation, striatal degeneration, and motor deficits[20]. This model has been widely used as a primary screening model for HD drug candidates[21]. Because CHIR99021 passes the blood–brain barrier[22], we intraperitoneally (i.p.) treated HD R6/2 and WT mice with the molecule (10 mg/kg/day, 5 days/week) starting at 6 weeks old (Fig. 2a) to determine whether CHIR99021 administration prevented rapid and severe progression of HD-related pathology. A 6-week treatment with CHIR99021 improved the survival of HD R6/2 mice when compared with vehicle-treated counterparts (Fig. 2b). CHIR99021 administration significantly reduced deficits in mouse locomotor activity as measured by an open-field activity chamber (Fig. 2c and Supplementary Fig. 2a) and attenuated the clasping behavior of HD R6/2 mice, which reflect motor deficits[23] (Fig. 2d). Moreover, the 6-week treatment regimen increased the immunodensity and protein levels of DARPP-32 (Fig. 2e and Supplementary Fig. 2d)—an MSN marker—and also reduced mtHtt aggregates (Fig. 2f and Supplementary Fig. 2b, c), a hallmark of HD pathology, in the striatum of HD R6/2 mice when compared with vehicle-treated HD mice.

Additionally, protein levels of postsynaptic density 95 (PSD95) and BDNF, which are decreased in HD animal models and human HD brains[24,25], were significantly enhanced following CHIR99021 administration (Supplementary Fig. 2d). These data suggested a protective effect of CHIR99021 in HD R6/2 mice. Importantly, this dosing regimen was not toxic: CHIR99021 treatments had no effects on behavioral status, body weight, and survival rates of WT mice over the 6-week treatment period (Fig. 2 and Supplementary Fig. 2).

We further assessed the effects of CHIR99021 in YAC128 mice, which express full-length human mtHtt and exhibit progressive motor abnormalities and late-stage selective striatal neuronal loss[26]. Specifically, YAC128 mice progressively develop HD-related symptoms starting from 6 months old, neurodegeneration at 9 months, and hypokinesis at 12 months[26]. Consequently, YAC128 mice more closely recapitulate the chronic neuropathology seen in human HD. Similar to a treatment regimen in previous work[27], we treated YAC128 mice and WT littermates with CHIR99021 and vehicle starting at 9 months old, 3 months after detectable motor deficits appear in YAC128 mice (Fig. 2g). We ended treatment at 12 months old. Initiating treatment at 9 months better mimicked post-symptomatic, chronic treatments administered to HD patients[28]. We assessed motor behaviors in mice once a month after the initial treatment. While rotarod latency in vehicle-treated YAC128 mice was dramatically decreased when compared to WT animals, CHIR99021 treatment significantly extended the duration of YAC128 mice running on the rotarod (Fig. 2h), suggesting an improvement in motor coordination activity in these animals. Additionally, total distance traveled and vertical activity in YAC128 mice were significantly improved after a 3-month treatment with CHIR99201 (Supplementary Fig. 2e). Consistent with the findings in HD R6/2 mice, the immunodensity and protein levels of DARPP-32 were increased and mtHtt aggregate numbers were attenuated in the striatum of CHIR99021-treated YAC128 mice when compared with vehicle-treated HD counterparts (Fig. 2i, j and Supplementary Fig. 2b, f, g). PSD95 and BDNF protein levels in YAC128 mouse striatum after CHIR99021 treatment were also restored to similar levels seen in WT littermates (Supplementary Fig. 2g). Notably, CHIR99021 treatment exerted no effects on YAC128 mice body weight, over a wide age range (Supplementary Fig. 2h). These results collectively demonstrated that CHIR99021 treatment reduced neuropathology and behavioral phenotypes in HD animal models.

**CHIR99021 acted via targets other than GSK3 in HD models.** CHIR99021 is widely used as a pan-GSK3 kinase inhibitor for studying glycogen metabolism, stem cell differentiation/proliferation,

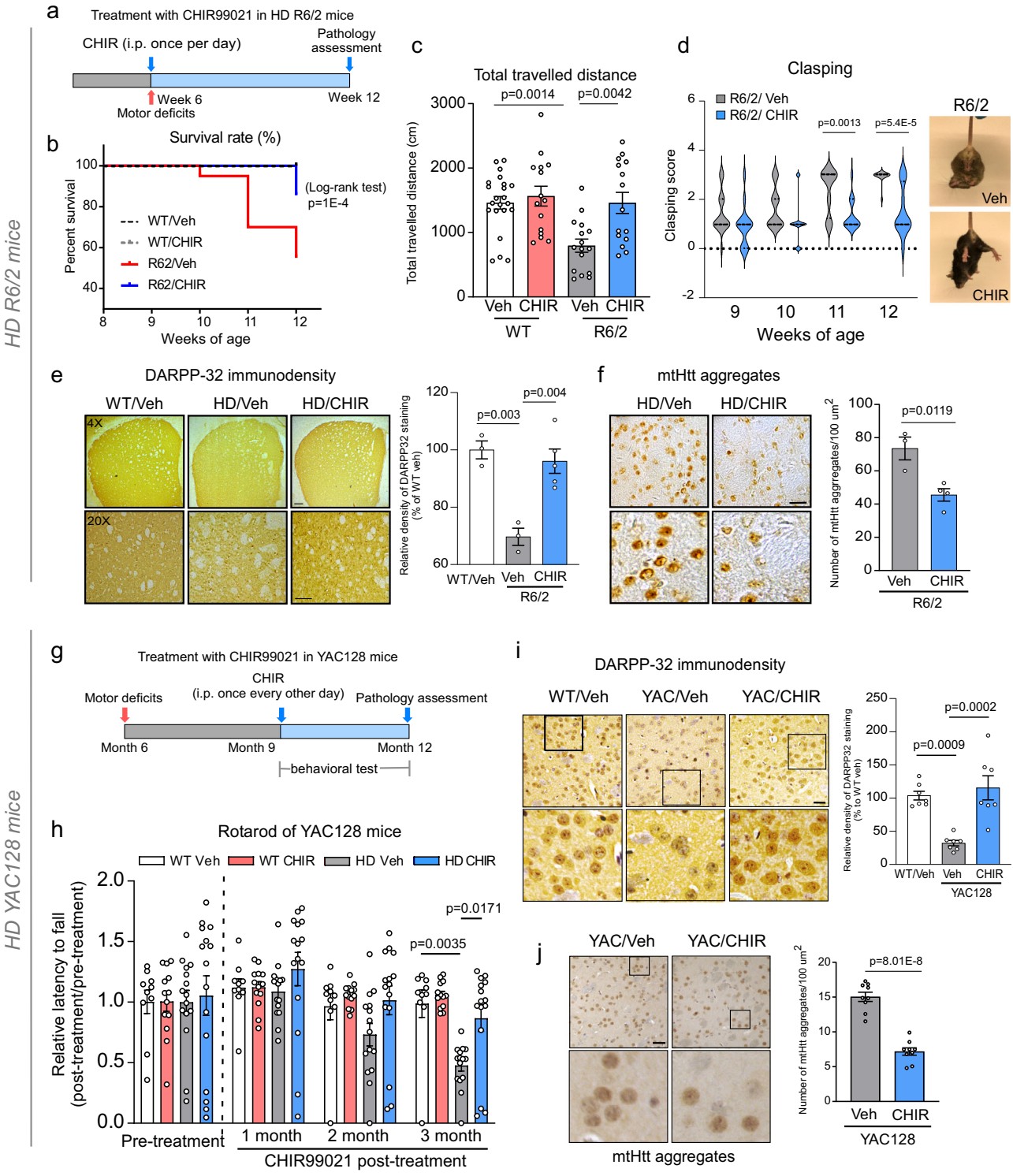

and neurogenesis[29,30]. However in HD, the two GSK3 isoforms, GSK3α and GSK3β, are either downregulated or inactivated by auto-inhibitory phosphorylation signals[31]. Consistent with previous work[31], we measured GSK3α/β protein levels in our HD cell culture and HD animal brains and observed a trend toward GSK3 downregulation (Supplementary Fig. 3a); levels of the GSK3 active forms (pY279-GSK3α and pY216-GSK3β) in HdhQ111 striatal cells and HD mouse brains were clearly lower than WT counterparts (Supplementary Fig. 3a). In addition, lithium chloride, a mood-stabilizing drug hypothesized to indirectly inhibit GSK3, lacked efficacy in the HD R6/2 mouse

model[32], casting further doubt on GSK3 as a therapeutic target in HD.

We conducted additional genetic and chemical–genetic studies to test our hypothesis that GSK3 is unlikely to be a functional target of CHIR99021 in HD models. We first downregulated GSK3 in HD cell culture models and assessed mitochondrial activity and cell survival. GSK3α/β knockdown using small interfering RNA (siRNA; Supplementary Fig. 3b) was insufficient to enhance MMP or prevent serum starvation-induced cell death in HdhQ111 striatal cells (Fig. 3a, b). In the presence of GSK3 siRNA, CHIR99021 retained its ability to enhance MMP

**Fig. 2 CHIR99021 is protective in HD animal models. a** Timeline of CHIR99021 treatment in R6/2 mice. CHIR99021 was intraperitoneally injected from 6 to 12 weeks old. **b** Survival of R6/2 mice from 6 to 12 weeks old was analyzed by the Log-rank (Mantel–Cox) test (two-tailed). **c** One hour of movement activity of R6/2 and WT littermates was determined by open-field activity chamber at 9 weeks old. Total traveled distance is shown. WT Veh $n = 22$, WT CHIR $n = 15$, R6/2 Veh $n = 16$, R6/2 CHIR $n = 15$ mice/group. **d** Hindlimb clasping activity was assessed using the tail suspension test once a week from 9 to 12 weeks old. $n = 10$–15 mice/group. **e** DARPP-32 immunodensity was examined in the dorsolateral striatum of mice. Scale bar = 100 μm. WT Veh $n = 3$, R6/2 Veh $n = 3$, R6/2 CHIR99021 $n = 5$ mice/group. **f** Brain sections were stained with anti-Htt antibodies (clone EM48), and mtHtt aggregate numbers were quantified. Scale bar = 50 μm. R6/2 Veh $n = 3$, R6/2 CHIR99021 $n = 4$ mice/group. **g** Timeline of CHIR99021 treatment in YAC128 mice. CHIR99021 was intraperitoneally injected from 9 to 12 months old. **h** Rotarod performance of mice was assessed at the indicated ages. WT Veh $n = 10$, WT CHIR $n = 13$, YAC128 Veh $n = 15$, YAC128 CHIR $n = 15$ mice/group. **i** DARPP-32 immunodensity in the striatum of 12-month-old YAC128 and WT mice. Left: representative image. Scale bar = 50 μm. Right: histogram of relative DARPP-32 immunodensity. $n = 7$ mice/group. **j** Quantification of mtHtt aggregates in YAC128 and WT mice. Left: representative images, scale bar = 50 μm. Right: quantification of mtHtt aggregates. $n = 9$ mice/group. All values are reported as mean ± SEM. Data are representative of at least three independent experiments. Data were compared using one-way ANOVA with Tukey's post hoc test in **c**, **e**, **h**, **i**, and the unpaired Student's $t$ test (two-tailed) in **d**, **f**, **j**. Exact $p$ values are shown in the figures.

and prevent cell death (Fig. 3a, b). Furthermore, we knocked out (KO) *GSK3B* in Neuro2a cells using CRISPR/cas9 (Supplementary Fig. 3c). Consistent with previous findings[33], overexpression of full-length Htt carrying 73 polyQ repeats (Myc-Htt-73Q, mtHtt) in Neuro2a cells caused mitochondrial depolarization and sensitized cells to oxidative stress. However, mitochondrial depolarization and oxidative stress-induced cell death were comparable in *GSK3B* WT and KO Neuro2a cells expressing Myc-Htt-73Q (Fig. 3c–e). CHIR99021 treatment in *GSK3B* KO Neuro2a cells expressing Myc-Htt-73Q improved MMP and reduced oxidative stress-induced cell death when compared to vehicle-treated cells (Fig. 3c–e). In parallel, we assessed a collection of 12 established GSK3 inhibitors spanning structurally diverse scaffolds across a wide concentration range. In addition to CHIR99021, only its close structural analog, CHIR98014, and the structurally distinct AZD1080 improved MMP and cell viability in HD cells (Fig. 3f). Recent work identified a series of GSK3 inhibitors, typified by BRD3731, that have greatly improved kinome-wide selectivity for GSK3 relative to CHIR99021 and other established inhibitors[34]. We demonstrated that these highly selective GSK3 inhibitors did not enhance MMP and cell viability in HdhQ111 striatal cells (Supplementary Fig. 3d). This genetic and chemical–genetic evidence suggested that CHIR99021 likely acted on protein targets other than GSK3α/β in HD.

**CHIR99021 modulated calpain and CAST levels in HD models.** To investigate mechanisms whereby CHIR99021 mediated mitochondrial protection and neuroprotection in HD models, we performed an unbiased label-free proteomics analysis of HdhQ111 cells treated with either CHIR99021 or BRD3731, which had no effect on MMP or cell viability (Supplementary Fig. 3d). Among the 1665 proteins identified in CHIR99021-treated HdhQ111 striatal cells, we focused on 282 altered proteins (i.e., >2-fold downregulated or upregulated relative to HdhQ7 cells) and simultaneously modified by CHIR99021 treatment (i.e., >2-fold upregulated or downregulated relative to HdhQ111 cells treated with DMSO; Fig. 4a). We removed proteins whose levels were altered by BRD3731 treatment to focus on alterations independent of GSK3 kinase inhibition. In total, 109 proteins meeting these criteria were subsequently used for Gene Ontology (GO) term enrichment analysis (Fig. 4a, marked with an asterisk (*)). A graphical comparison of enriched biological pathways showed significant alterations in protein degradation pathways, in which proteins enriched in proteolysis were mostly affected (Fig. 4b). Among proteins in the "proteolysis" GO class, the cysteine protease calpain I was strikingly upregulated in HdhQ111 striatal cells relative to WT HdhQ7 cells and was restored in CHIR99021-treated HdhQ111 cells (Fig. 4b). Interestingly, CAST, a highly specific endogenous calpain inhibitor[35], was inversely regulated by CHIR99021 treatment, i.e., CAST

**Table 1 HD patient and normal subject postmortem brain samples (frozen).**

| ID | Brain region | Diagnosis |
|---|---|---|
| 5214 | Caudate putamen | Control |
| 5222 | Caudate putamen | Control |
| 5256 | Caudate putamen | Control |
| 2903 | Caudate putamen | HD |
| 4518 | Caudate putamen | HD |
| 2869 | Caudate putamen | HD |
| 3573 | Caudate putamen | HD |
| 4132 | Caudate putamen | HD |
| 4283 | Caudate putamen | HD |

protein levels were lower in HdhQ111 cells relative to HdhQ7 cells and were increased by CHIR99021 treatment in HdhQ111 cells (Fig. 4b). Notably, BRD3731 treatment exerted no effects on calpain I or CAST levels in HdhQ111 cells (Fig. 4b). This unbiased proteomics analysis suggested that CHIR99021 treatment restored calpain I and CAST proteins to WT levels. Since calpain activation was previously shown to exacerbate neurodegeneration[36], and CAST was an exclusively specific endogenous calpain inhibitor, our findings led us to consider CAST-mediated suppression of calpain activity as a functional target of CHIR99021 in HD.

We first determined whether decreased CAST protein levels occurred in human HD, therefore we examined CAST protein expression in HD patient post-mortem brains (Table 1). The protein levels of CAST were reduced by approximately 70% in the caudate putamen of patients with HD when compared with normal subjects (NS; Fig. 4c). In contrast, levels of the cytosolic protein, enolase, were comparable in postmortem caudate putamen of patients with HD and NS (Fig. 4c), suggesting a selective decrease in CAST. Levels of cleaved α-spectrin, a known calpain substrate, were significantly increased in HD patient brains relative to NS brains (Fig. 4c) reflecting calpain activation. CAST protein levels in HD patient-derived fibroblasts were consistently lower than that in control fibroblasts but were significantly elevated by CHIR99021 treatment (Supplementary Fig. 4a). CHIR99021 treatment exerted no effects on CAST protein levels in normal fibroblasts.

The genetic upregulation of CAST was previously shown to alleviate HD phenotypes in mice[37]. We next assessed whether CHIR99021 also affected CAST protein levels in HD cell cultures and HD mice. We observed a marked decrease in CAST levels in HdhQ111 striatal cells, HdhQ7 striatal cells exposed to 3-nitropropionic acid (3-NP; a mitochondrial toxin that causes HD-like symptoms in rodents and primates[38]), and HEK293 cells overexpressing mtHtt carrying either 73Q or 145Q when

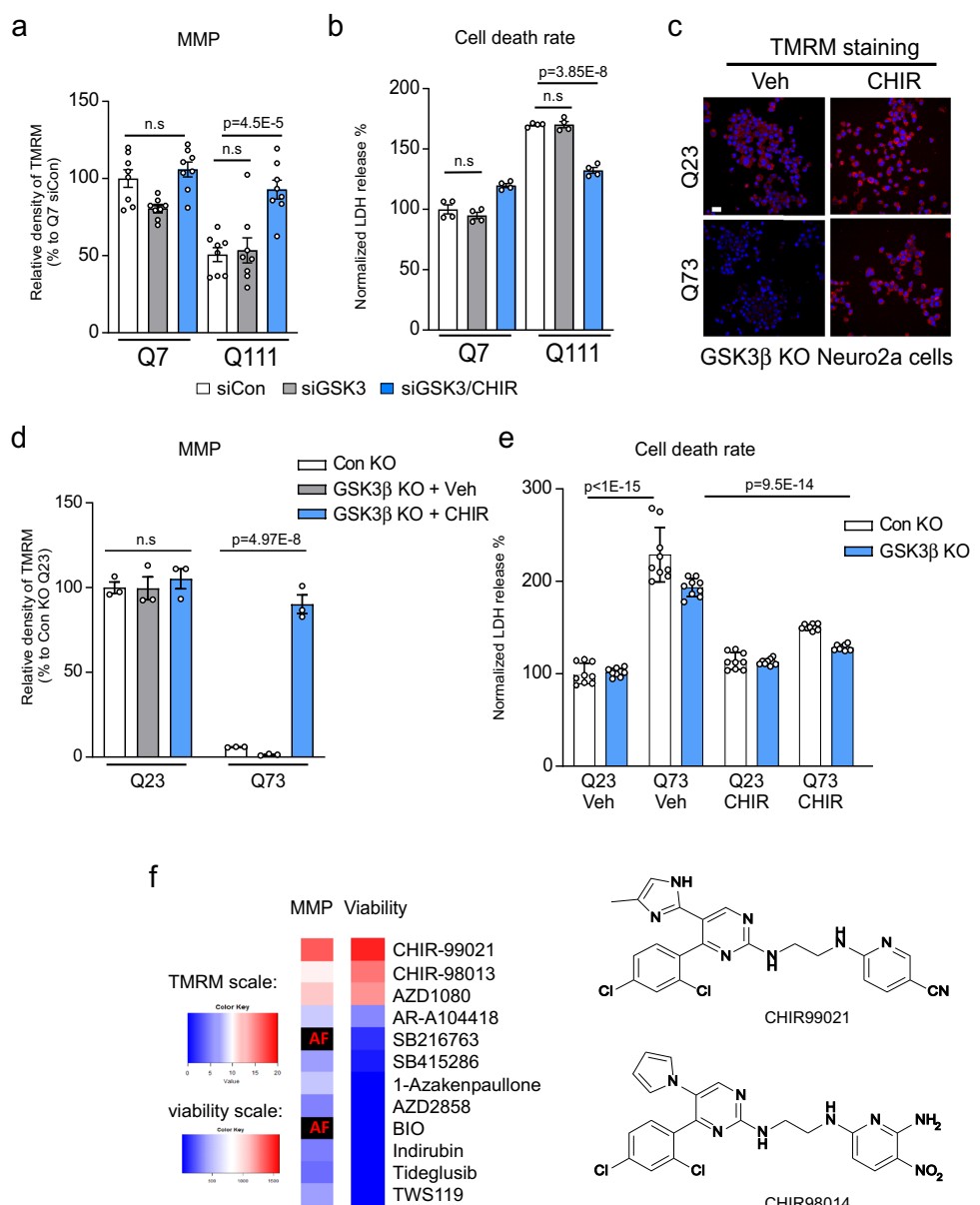

**Fig. 3 The protective effects of CHIR99021 in HD is independent of GSK3.** HdhQ7 and HdhQ111 cells were transfected with small interfering RNA (siRNA) control (siCon) or GSK3 (siGSK3) for 3 days, followed by CHIR99021 (3 μM) treatment for 48 h. **a** Mitochondrial membrane potential (MMP) was examined by TMRM staining; quantification of TMRM density is shown ($n = 8$). **b** Cell death was measured by lactate dehydrogenase (LDH) release after 16 h serum starvation ($n = 4$). Neuro2a cells were transfected with vectors expressing control guide RNA (gRNA) or GSK3β gRNA and Cas9. After confirming GSK3 knockout (Supplementary Fig. 3c), cells were transfected with Myc-tagged Q23-Htt (Q23) or Q73-Htt (Q73) constructs for 12 h, followed by a 2-day incubation with CHIR99021 (3 μM) or vehicle (Veh). **c** Representative TMRM images of the indicated groups. Scale bar = 30 μm. **d** Quantification of TMRM fluorescence density is shown ($n = 3$). **e** Cell death in the indicated groups was measured by LDH release ($n = 9$). **f** Heat maps representing the effects of 12 GSK3 inhibitors on MMP (left) and cell viability following serum withdrawal (right). All molecules were assessed at 8 doses ranging from 0.078 to 10 μM. MMP data are presented as the area under the dose–response curve of TMRM signal increase relative to DMSO-treated cells. Viability data are presented as the integral of the increase in non-apoptotic cells across all doses. AF: compound autofluorescence prevents an accurate assessment. TMRM scale: 0–20, viability scale: 0–1500. The CHIR99021 structure and its chemical analog, CHIR98014, are shown on the right. All values are reported as mean ± SEM and compared using one-way ANOVA with Tukey's post hoc test in **a**, **b**, two-way ANOVA with Šídák's post hoc test in **d**, and two-way ANOVA with Dunnett's post hoc test in **e**. Data are representative of at least three independent experiments. Exact $p$ values are shown in the figures.

compared with the control groups (Supplementary Fig. 4b–d). CHIR99021 treatment corrected decreased CAST protein levels in these HD cell cultures (Supplementary Fig. 4b–d), supporting our earlier proteomics data (Fig. 4a, b).

In the striatum of vehicle-treated R6/2 mice, we confirmed that CAST protein levels were decreased, whereas calpain I protein levels were increased when compared with vehicle-treated WT littermates (Fig. 4d). Alpha-spectrin and p35 are known calpain substrates[39,40]. Levels of cleaved α-spectrin and p25, a cleaved product of p35, were increased in vehicle-treated R6/2 mice relative to vehicle-treated WT animals (Fig. 4d and Supplementary Fig. 4e), confirming calpain activation in HD mice. A 6-week administration of CHIR99021 corrected CAST protein levels and abrogated increased calpain I, p25, and cleaved α-spectrin levels

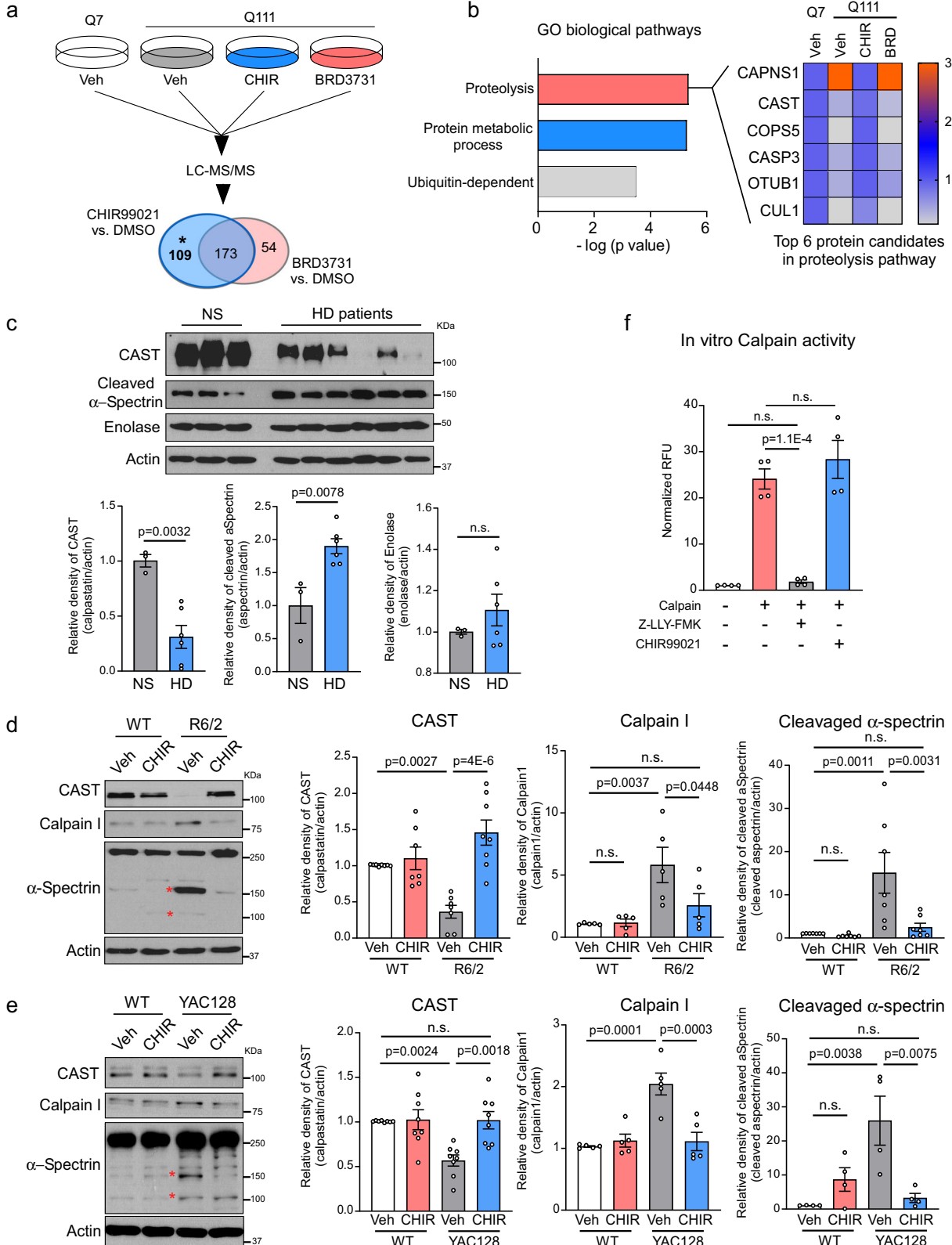

in R6/2 mice striatum (Fig. 4d and Supplementary Fig. 4e). Similarly, CAST protein levels were decreased and calpain I, p25, and cleaved α-spectrin protein levels increased in vehicle-treated YAC128 mice striatum when compared with WT littermates, which were all corrected by CHIR99021 treatment (Fig. 4e and Supplementary Fig. 4e). No effects of CHIR99021 on these protein levels were observed in WT mice (Fig. 4d, e and

Supplementary Fig. 4e). Additionally, we directly measured calpain activities in the brains of R6/2 and YAC128 mice. As predicted by upregulated calpain I protein levels observed earlier, calpain enzymatic activity was substantially higher in both R6/2 and YAC128 mice (Supplementary Fig. 4f, g). CHIR99021 treatment diminished calpain activity to near WT levels (Supplementary Fig. 4f, g), consistent with its ability to abrogate

**Fig. 4 CHIR99021 treatment restores the CAST-calpain pathway in HD models. a** HdhQ111 cells were treated with vehicle (Veh; DMSO), CHIR99021 or BRD3731 (3 μM each) for 2 days. Total cell lysates from HdhQ111 and HdhQ7 cells were harvested and label-free tandem mass spectrometry (LC-MS/MS) analyses performed. The numbers of changed proteins in CHIR99021- or BRD3731-treated cells are shown. The 173 proteins altered by BRD3731 were removed from total altered proteins in CHIR99021-treated HdhQ111 striatal cells. **b** The biological functions of the 109 proteins from the pool marked asterisk (*) in **a** were analyzed. Heat map analysis of enriched proteins in proteolysis pathways. Data are analyzed by one-sided Fisher's exact test. **c** Total protein lysates were harvested from postmortem caudate putamen of normal subjects (NS) ($n = 3$) and patients with HD ($n = 6$) and subjected to western blotting. Normal subject and patient information is listed in Table 1. Quantification of the relative abundance of CAST, cleaved-α-spectrin, and enolase is shown. **d** Total protein lysates were harvested from the striatum of 12-week-old R6/2 and WT littermates treated with vehicle or CHIR99021 and subjected to western blotting with the indicated antibodies. *Cleaved α-spectrin. Histograms show the quantification of relative calpain ($n = 5$), CAST ($n = 7$), and cleaved α-spectrin ($n = 7$) levels in the indicated groups. **e** Total protein lysates were harvested from the striatum of YAC128 and WT littermates treated with vehicle or CHIR99021 and subjected to western blotting with the indicated antibodies. Histograms show the quantification of relative calpain ($n = 5$), CAST ($n = 8$), and cleaved α-spectrin ($n = 4$) levels in the indicated groups. **f** In vitro calpain activity was assayed in the presence or absence of CHIR99021 ($n = 4$). Z-LLY-PMK was used as a selective calpain inhibitor. All values are reported as mean ± SEM and compared using one-way ANOVA with Tukey's post hoc test in **f**, Dunnett's post hoc test in **d**, **e**, and the unpaired Student's $t$ test (two-tailed) in **c**. Data are representative of at least three independent experiments. Exact p values are shown in the figures.

calpain substrate cleavage. However, in an in vitro calpain enzymatic activity assay, the addition of CHIR99021 had no effect on the proteolytic activity of calpain, even though the selective calpain inhibitor, Z-LLY-FMK, abolished calpain activity (Fig. 4f). Collectively, our data from several human and mouse HD models demonstrated a consistent modulation of CAST protein levels by CHIR99021 treatment, thereby suppressing calpain activation.

The truncated p35 form resulting from calpain activation, p25, is a specific and constitutive activator of cyclin-dependent kinase 5 (CDK5) via kinase binding. CDK5 overactivation causes neuronal death in many neurodegenerative diseases, including HD[41]. We observed that CDK5 knockdown by siRNA in HdhQ111 striatal cells improved MMP and serum starvation-induced cell death. Treatment with CHIR99021 showed no additional protective effects on MMP and cell viability in the absence of CDK5 (Supplementary Fig. 4h). Notably, we confirmed data from a previous report[42] that CHIR99021 only weakly inhibited CDK5 enzymatic activity in vitro (IC$_{50}$ = 5.58 μM) (Supplementary Fig. 4i), suggesting that the effects of CHIR99021 on CDK5 activation in HD cells were indirect. These findings further supported our observations that the mitochondrial protection effects afforded by CHIR99021 in HD models were dependent on a CAST–calpain cascade, upstream of CDK5.

**CHIR99021 stabilized CAST to improve mitochondrial function and reduce neuronal death in HD models.** To determine whether the protective effects of CHIR99021 in our HD models depended on the presence of CAST, we knocked down CAST using short hairpin RNA (shRNA) in HdhQ111 cells (Fig. 5a) and determined MMP and cell viability (Fig. 5b, c). In the presence of the control shRNA (shCon), CHIR99021 greatly improved MMP and cell viability in these cells. In contrast, CAST knock-down abolished the protective effects of CHIR99021 toward MMP and cell viability (Fig. 5b, c). In parallel, we also knocked down calpain I by shRNA (shCAPN1) (Fig. 5a). This knock-down significantly improved MMP and reduced cell death in HdhQ111 cells (Fig. 5d, e). Treatment with CHIR99021 had no additional effects on MMP and cell viability in HdhQ111 cells expressing shCAPN1; the effects were similar to calpain knock-down or CHIR99021 treatment-alone effects (Fig. 5d, e). In an additional pharmacological strategy to suppress calpain activity, we treated HdhQ111 cells with a CAST mimicking peptide, N-acetyl-calpastatin, which has an inhibitory domain targeting calpain[43]. Again, treatment with N-acetyl-calpastatin significantly improved MMP and reduced cell death in HdhQ111 cells (Supplementary Fig. 5a, b). Also, co-treatment with CHIR99021 and N-acetyl-calpastatin had no additional effects on MMP and cell viability (Supplementary Fig. 5a, b).

Next, to examine whether CAST mediated the neuronal protection of CHIR99021 in human HD, we downregulated CAST using lentiviral shRNA in neurons derived from HD patient iPSCs (Fig. 5f). Consistent with our earlier findings (Fig. 1f, g), CHIR99021 treatment significantly improved MMP and cell viability in HD patient iPS cells-derived neurons infected with control shRNA (Fig. 5g). However, in patient iPS cells-derived neurons infected with CAST shRNA (shCAST), CHIR99021 had no protective effects on MMP or cell viability (Fig. 5g). Moreover, CHIR99021 treatment failed to rescue the dendritic and axonal shortening in shCAST-expressing neurons derived from HD patient iPS cells (Fig. 5h–j). Collectively, our results indicated that CHIR99021-mediated mitochondrial protection in HD cells requires the presence of CAST.

We next investigated how CHIR99021 enhanced CAST protein levels in our HD models. We observed that CAST and CAPN1 mRNA levels were comparable in vehicle- and CHIR99021-treated WT and HD mice (Supplementary Fig. 5c, d), suggesting transcriptionally independent effects. Treatment with an autophagy inhibitor bafilomycin A (BFA) did not affect CAST levels in either HdhQ7 or HdhQ111 cells (Supplementary Fig. 6a), thereby excluding the possibility of autophagy-mediated protein loss. We then asked whether a decreased CAST protein level results from protein degradation in the context of HD. To answer this, we treated HdhQ7 and HdhQ111 striatal cells with a protein synthesis inhibitor, cycloheximide (CHX), and determined CAST protein levels. Western blotting showed that CAST protein levels were diminished in a time-dependent manner after CHX addition to HdhQ111 cells, but not HdhQ7 cells (Supplementary Fig. 6b). The loss of CAST in CHX-treated HdhQ111 cells was prevented by the addition of multiple structurally unrelated proteasome inhibitors, including lactacystin (Lac), epoxomicin (epo), or MG132 (Supplementary Fig. 6c–e). Interestingly, while these additions restored CAST protein levels in HdhQ111 cells, co-treatment with CHIR99021 did not generate additional improvements under the same conditions (Fig. 6a, b and Supplementary Fig. 6f). Also, CHIR99021 treatment reduced calpain I protein levels and suppressed its activity in HdhQ111 cells in the presence of these inhibitors (Fig. 6a–c and Supplementary Fig. 6f, g). In addition, treatment with MG132 induced a substantial accumulation of ubiquitinated CAST in HdhQ111 striatal cells, which was abolished by CHIR99021 treatment (Fig. 6d). These results collectively suggested that CAST had a shorter half-life in HD cells than in WT cells; CAST was degraded via the UPS, with CAST ubiquitination and degradation both prevented by CHIR99021. CHIR99021 treatment had no effect on proteasome activity in vitro and in HdhQ7/111 cells (Supplementary Fig. 6h, i) nor on MCL1 protein levels, a well-known proteasome substrate[44] (Fig. 6a, b and

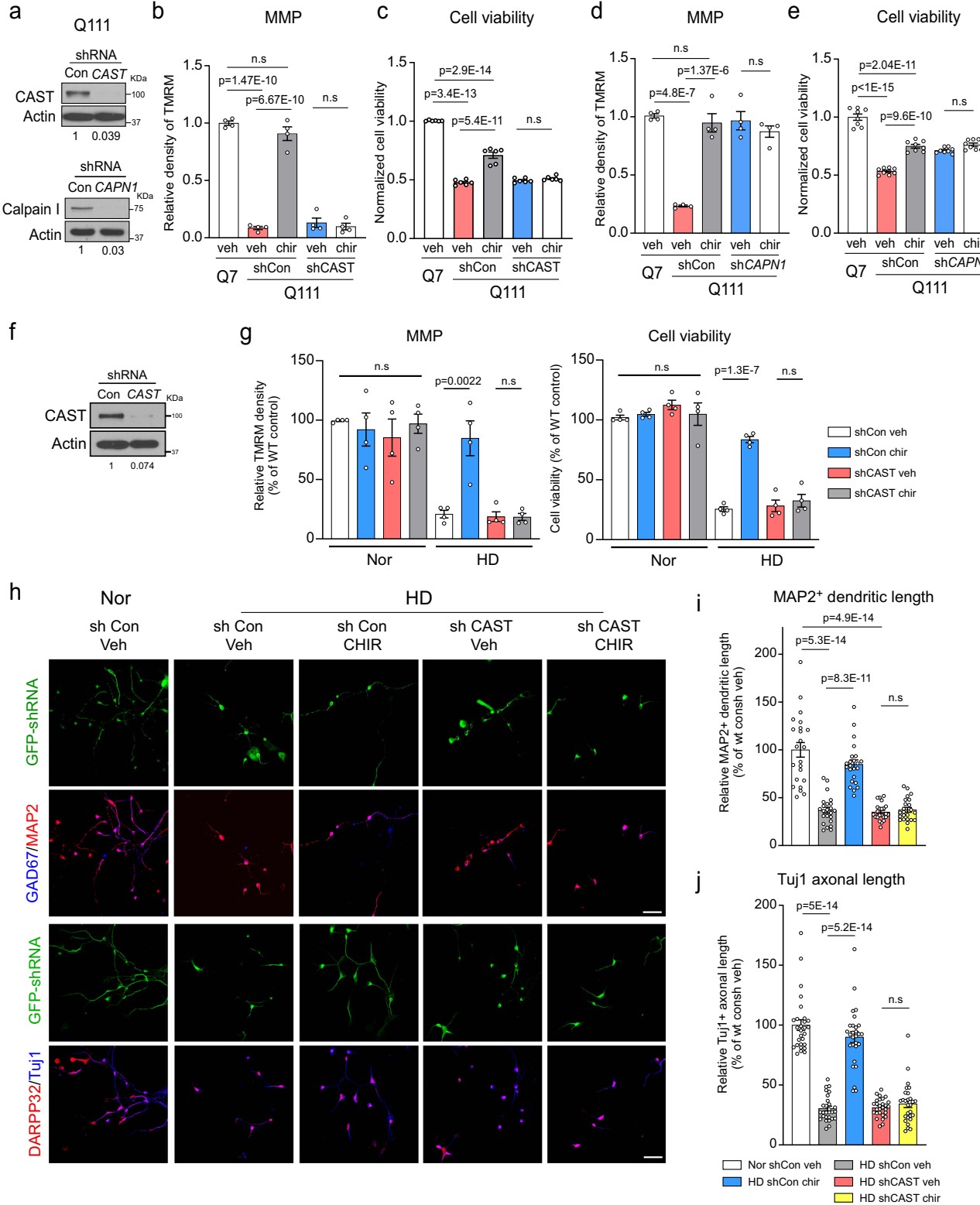

Supplementary Fig. 6f), thereby excluding a global inhibition of CHIR99021 on proteasome activity. Taken together, our results led us to propose a model that CHIR99021 treatment prevents the UPS-mediated degradation of CAST in the context of HD.

**CHIR99021 inhibited Drp1 hyperactivation downstream of CAST in HD models**. Drp1 is a primary mitochondrial fission protein activated in various HD models[10,11]. Its hyperactivation causes mitochondrial fragmentation, leading to wide-ranging mitochondrial damage, including mitochondrial depolarization, oxidative stress, and bioenergetic defects, which eventually lead to neurodegeneration. We and others showed that the inhibition of Drp1 hyperactivation via genetic or pharmacological manipulation protected mitochondrial and neuronal degeneration, which in turn reduced HD-associated neuropathology[10,11,14]. Thus, we

**Fig. 5 CHIR99021 neuroprotection in HD patient neurons requires CAST. a** Knock-down efficiency of CAST and calpain I (*CAPN1*) short hairpin RNA (shRNA) was confirmed by western blotting. **b** Mitochondrial membrane potential (MMP) was assessed using a TMRM fluorescence probe in control shRNA (shCon) or CAST shRNA (shCAST) expressing cells treated with CHIR99021 (chir) or vehicle (veh) (*n* = 4). **c** Cell viability was assessed by MTT assay in shCon- or shCAST-expressing cells treated with CHIR99021 or vehicle (*n* = 6). **d** MMP was measured by TMRM staining in control shRNA (shCon)- or sh*CAPN1*-expressing cells treated with CHIR99021 or vehicle (*n* = 4). **e** Cell viability was measured by MTT assay in shCon or sh*CAPN1* cells treated with CHIR99021 or vehicle (*n* = 8). Mixed striatal neurons were differentiated from iPS cells of patients with HD and normal subjects. Twenty days after neuronal differentiation, neurons were infected with lentivirus expressing GFP-labeled shCon or shCAST. **f** Knock-down efficiency of shCAST in neurons derived from HD patient iPS cells was confirmed by western blotting. Two days after infection, neurons derived from iPS cells from normal subjects (Nor) and HD patients (HD) were treated with CHIR99021 at 1 μM for 5 consecutive days. **g** MMP was assessed using a TMRM fluorescence probe (*n* = 4). Cell viability was measured by MTT assay after brain-derived neurotrophic factor (BDNF) withdrawal for 12 h (*n* = 4). **h** Neurons were stained with anti-MAP2 (red)/anti-GAD67 (blue) or anti-β-tubulin (clone Tuj1) (blue)/anti-DARPP-32 (red) to indicate dendritic and axonal morphology, respectively. Scale bar = 30 μm. **i** Quantification of MAP2+ neuronal dendrite length (*n* = 23 cells/group). **j** Quantification of Tuj1+ neuronal axon length (*n* = 28 cell/group). All values are reported as mean ± SEM and compared using one-way ANOVA with Tukey's post hoc test. Data are representative of at least three independent experiments. Exact *p* values are shown in the figures.

investigated whether CHIR99021 improved mitochondrial function by protecting Drp1-mediated mitochondrial fragmentation in HD. Consistent with our previous observations[14,18], HdhQ111 striatal cells exhibited extensive mitochondrial fragmentation relative to HdhQ7 cells (Fig. 7a). In contrast, CHIR99021 treatment reduced the percentage of HdhQ111 cells exhibiting fragmented mitochondria, whereas treatment exerted minor effects on mitochondrial morphology in HdhQ7 cells (Fig. 7a). Similarly, CHIR99021 treatment greatly reduced mitochondrial fragmentation in fibroblasts from HD patients and exerted no effects in control fibroblasts (Supplementary Fig. 7a). By examining protein alterations implicated in mitochondrial quality control pathways, including mitochondrial dynamics, biogenesis, and autophagy (Fig. 7b), we observed that CHIR99021 treatment mainly reduced phosphor-Ser616 Drp1 (Drp1-pS616) levels, a marker of Drp1 activation, but had no effects on other proteins relevant to mitochondrial quality control examined in HdhQ111 cells (Fig. 7b and Supplementary Fig. 7b). In contrast, treatment with BRD3731 elicited no effects on Drp1-pS616 levels (Supplementary Fig. 7c), further demonstrating that the protective effects of CHIR99021 in HD cells were independent of GSK3 inhibition. In addition, CHIR99021 did not affect the endoplasmic reticulum (ER)-stress-related proteins, CCAAT-enhancer-binding protein homologous (CHOP) and GRP78, upon thapsigargin or tunicamycin treatment, suggesting a selective regulation toward mitochondria (Supplementary Fig. 7d). We further confirmed Drp1 activation in our HD cells and HD mouse brains, as evidenced by Drp1 mitochondrial translocation and Drp1 oligomerization[45]. CHIR99021 treatment significantly reduced the level of mitochondrial Drp1 in HdhQ111 cells and in the striatum of HD R6/2 mice and abolished the accumulation of Drp1 oligomers (Fig. 7c and Supplementary Fig. 7e, f). These findings suggested Drp1-mediated mitochondrial fragmentation as a molecular pathway by which CHIR99021 mediates mitochondrial protection.

Next, we determined whether the protective effects of CHIR99021 on Drp1-mediated mitochondrial fragmentation in HD cells were dependent on CAST expression. The administration of 3-NP to HdhQ7 cells caused the mitochondrial translocation of Drp1, which was abolished upon CHIR99021 treatment (Fig. 7d). In CAST-deficient cells, CHIR99021 lost its ability to block Drp1 mitochondrial accumulation induced by 3-NP (Fig. 7d). As a result, CHIR99021 treatment exerted no effects on mitochondrial fragmentation induced by 3-NP in CAST-deficient HdhQ7 cells (Supplementary Fig. 7g). Furthermore, CHIR99021 treatment failed to reduce mitochondrial fragmentation in HdhQ111 striatal cells in the presence of CAST shRNA (Fig. 7e, f) or calpain shRNA (Fig. 7g, h). In parallel, CAST stabilization by treating cells with *N*-acetyl-calpastatin

similarly reduced the percentage of HdhQ111 cells with fragmented mitochondria relative to HdhQ7 cells. Co-treatment with CHIR99021 and *N*-acetyl-calpastatin had no additional impact on correcting mitochondrial morphology in HdhQ111 cells (Supplementary Fig. 7h). These results demonstrated that CAST was upstream of Drp1-mediated mitochondrial fragmentation in HD. Our findings also suggested the existence of a CAST–calpain–Drp1 signaling axis as a mechanism underpinning CHIR99021 actions in HD.

## Discussion

In this study, by focusing on mitochondrial defects in HD, we identified several findings (Fig. 7i). First, using a combination of HTS assays and in vitro and in vivo experimental HD models, we identified CHIR99021 as an enhancer of mitochondrial function: it improved MMP, mitochondrial morphology, mitochondrial bioenergetics, and, ultimately, neuronal survival in several HD cell culture models. Second, we demonstrated a strong efficacy for CHIR99021 treatment in two HD mouse models and neurons derived from HD patient iPS cells, thereby supporting our hypothesis that mitochondrial enhancers effectively reduced HD-associated neuropathology. Third, we discovered a mechanism of action for CHIR99021 in HD, independent of its canonical GSK3 inhibitory effects. We determined that the CAST–calpain–Drp1 signaling axis mediated the protective role of CHIR99021, providing insights into HD pathogenesis. Therefore, our findings not only identified a druggable signaling axis but also suggested small molecules such as CHIR99021 as potential therapeutic leads for HD.

CHIR99021 is a GSK3 kinase inhibitor functioning as a competitive inhibitor of ATP binding[46]. While it exhibits some selectivity for GSK3 versus other kinases, a recent kinome screening indicated that CHIR99021 can inhibit >20 other kinases, supporting its off-target effects[47]. Being a GSK inhibitor, CHIR99021 potentiates insulin activation of the glucose transporter and promotes replication and survival of pancreatic β-cells[29,48]. Given GSK3 also regulates Wnt signaling by directly phosphorylating β-catenin, CHIR99021 has been widely used as a mediator in stem-cell-related modeling, such as organoid generation and differentiation to central nervous system neurons from iPS cells[49]. However, in this study, our evidence excluded GSK3 as a functional target of CHIR99021 in HD models. Interestingly, Ma et al. recently showed that CHIR99021 induced long intergenic non-coding RNA expression via epigenetic manipulation and promoted mitochondrial biogenesis and oxidative capacity in human endodermal progenitor cells[50]. Thus, our findings and others support the notion that CHIR99021 acts on targets other than GSK3.

Calpain I is the major isoform of cysteine protease calpains in neurons and is typically maintained in an inactive precursor state

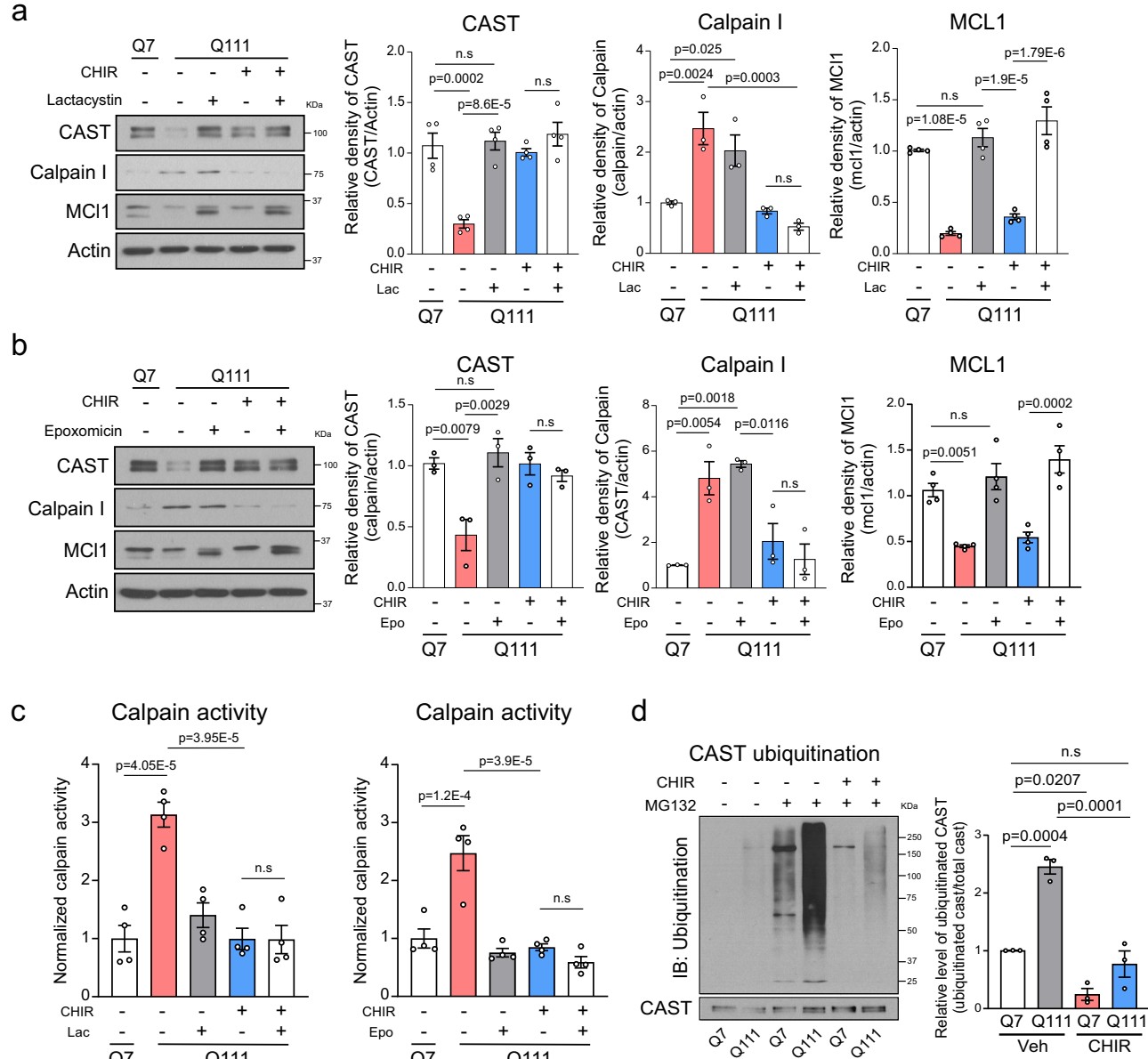

**Fig. 6 CHIR99021 treatment stabilizes CAST in HD models.** Total lysates from HdhQ7 and HdhQ111 cells were harvested after the indicated treatments; 3 μM CHIR99021 for 48 h followed by **a** 2.5 μM lactacystin (Lac) for 12 h or **b** 250 nM epoxomicin (Epo) for 12 h and subjected to western blotting with the indicated antibodies. Histograms show the relative abundance of CAST (n = 4), calpain I (n = 3), or MCL1 (n = 4). **c** Calpain enzymatic activity was examined in HdhQ7 and HdhQ111 cells with the indicated treatments: 48 h treatment with CHIR99021 followed by 12 h treatment with Lac (n = 4) or Epo (n = 4). **d** Total immunoprecipitation (IP) with an anti-CAST antibody followed by western blotting with an anti-ubiquitin antibody. Representative immunoblots and quantification of the relative abundance of ubiquitinated CAST (n = 3). All values are reported as mean ± SEM. Data are representative of at least three independent experiments. Data were compared using one-way ANOVA with Tukey's post hoc test. Exact p values are shown in the figures.

until activation by micro-molar calcium levels[51]. Under physiological conditions, calpain I limits cellular proteolysis and is tightly regulated by its endogenous proteinaceous calpain inhibitor, CAST[35]. In HD, calpains not only cleave mtHtt but also exhibit an overactivation in the molecular context of HD[52]. Moreover, the calpain-mediated conversion of the CDK5 activator p35 to p25 results in aberrant CDK5 activity, which contributes to selective striatal cell death in HD[41]. In addition, calpain cleaves DARPP-32, a key molecule that supports the survival of medium striatal neurons[53]. Thus, activated calpains represent important players in the molecular pathogenesis of HD. However, the development of inhibitors directly targeting calpains has proven challenging due to poor specificity and potential

toxicity[54]. As an alternative, the selective inhibition of calpain activity via elevation of its endogenous inhibitor, CAST, may be attractive. The *CAST* KO led to increased calpain activation, aggravated mtHtt aggregation, and worsened HD phenotypes[55]. In contrast, CAST overexpression in fly and mouse HD models ameliorated HD-associated neuropathology and behavioral deficits[37]. Importantly, the long-term elevation of CAST in HD mice did not result in any overtly deleterious phenotypes[37]. Thus, CAST manipulation rather than calpain inhibition may be a more effective strategy for developing HD therapeutic strategies. Consistent with previous reports, we observed a significant increase in calpain I expression and decrease in CAST expression in our HdhQ111 mutant striatal cells and two HD transgenic mouse

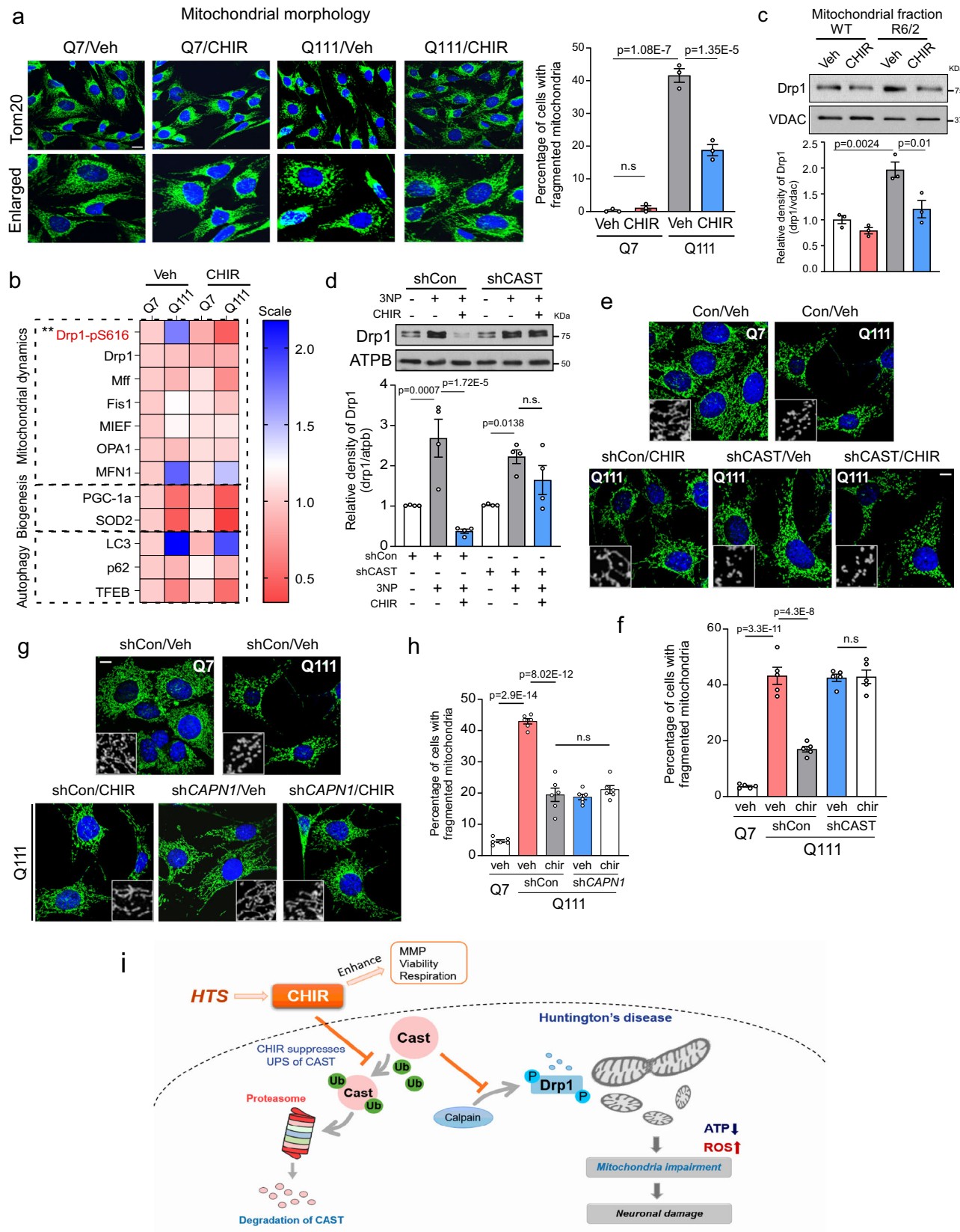

models relative to controls. CHIR99021 selectively elevated CAST protein levels by preventing its degradation. As a result, CHIR99021 treatment limited calpain activity and the subsequent cleavage of its substrates (p35 and α-spectrin), which in turn

reduced striatal neuronal loss. Calpain mediated the cleavage of mtHtt leading to Htt aggregation; calpain inhibition via CAST overexpression stimulated the autophagy-dependent clearance of mtHtt[37]. These lines of evidence also interpret our findings that

**Fig. 7 CAST-dependent Drp1-mediated mitochondrial fragmentation mediates CHIR99021 protection in HD. a** HdhQ7 and HdhQ111 cells were treated with DMSO (vehicle) or CHIR99021 (3 μM) for 2 days, with cells stained with anti-Tom20 antibody to assess mitochondrial morphology. Left: representative images. Scale bar = 30 μm. Right: histogram showing the percentage of cells with fragmented mitochondria. At least 100 cells/group were counted ($n = 3$). **b** Total levels of the indicated proteins classified into three pathways (mitochondrial dynamics, biogenesis, and autophagy) were examined and quantified. The mean relative protein level value was presented in a dual-color heat map. \*\*$p = 0.0055$, Q111-CHIR99021 versus Q111-Veh. Relative protein abundance scale: 0.5–2.0. **c** Mitochondria were isolated from the striatum of 12-week-old R6/2 and WT littermates treated with vehicle or CHIR99021 and subjected to western blotting. The relative abundance of mitochondrial Drp1 was quantitated (histogram). **d** Mitochondrial fractions were isolated from HdhQ7 cells expressing control shRNA (shCon) or CAST shRNA (shCAST) after treatment with 3-nitropropionic acid (3-NP) and/or CHIR99021. Representative immunoblots and quantification of Drp1 relative levels ($n = 4$). **e** Representative images of anti-Tom20 staining (green) in shCon- or shCAST-expressing cells treated with Veh or CHIR99021. Scale bar = 30 μm. **f** Histogram shows the percentage of cells with fragmented mitochondria. At least 100 cells/group were counted ($n = 5$). **g** Representative images of anti-Tom20 staining (green) in shCon- or calpain I shRNA (sh*CAPN1*)-expressing HdhQ111 cells treated with Veh or CHIR99021. Scale bar = 30 μm. **h** Histogram shows the percentage of cells with fragmented mitochondria. At least 100 cells/group were counted ($n = 6$). All values are reported as mean ± SEM and compared using one-way ANOVA with Tukey's post hoc test. Data are representative of at least three independent experiments. Exact $p$ values are shown in the figures. **i** Summary graphic depicting the ability of CHIR99021 to stabilize CAST and promote mitoprotection and neuroprotection.

CHIR99021 administration reduced mtHtt aggregates in the striatum of HD R6/2 and YAC128 mice. Notably, we observed a significant loss of CAST protein expression in HD patient post-mortem brains and HD patient fibroblasts, suggesting that decreased CAST is a pathological marker for HD. We also demonstrated that CHIR99021 treatment prevented the UPS-mediated degradation of CAST in HD and restored CAST loss in HD patient fibroblasts.

Our study reveals the UPS-mediated degradation of CAST. In HdhQ111 cells, the CHX-induced degradation of CAST was prevented by treating cells with selective proteasome inhibitors, suggesting a high turn-over rate for the CAST protein mediated by the proteasome. Although CHIR99021 treatment prevented CAST loss in HdhQ111 cells, it did not improve MCL1 protein levels—a protein degraded by the UPS pathway[44]—consistent with the observation that CHIR99021 did not affect proteasome activity. These lines of evidence suggest that CHIR99021 may selectively modulate CAST degradation rather than bulk protein degradation by the proteasome. CAST can be cleaved and degraded by certain proteases, including calpain[56]. However, our results excluded the possibility that CHIR99021 directly affected calpain enzymatic activity. Because the accumulation of CAST ubiquitination was abolished upon CHIR99021 treatment in HdhQ111 cells, we speculate that CHIR99021 might inhibit ubiquitin ligases or stimulate deubiquitinases that regulate CAST ubiquitination and degradation. Therefore, CHIR99021 may serve as a CAST stabilizer to suppress calpain-mediated cellular injury in HD. Future investigations into the regulation of CAST ubiquitination in HD are warranted to elucidate detailed pathways where CHIR99021 stabilizes CAST.

Calpain activation and CAST loss have been linked to mitochondrial dysfunction[57]. The presence of calpain and CAST on mitochondria has been well documented[58,59]. While calpain accumulation on mitochondria leads to mitochondrial depolarization, oxidative stress, and energy depletion[60], CAST over-expression attenuates calpain-mediated mitochondrial damage in response to various stimuli[61]. Mitochondria are dynamic organelles that are balanced by fusion and fission processes[62]. Enhanced fission and fragmentation result in mitochondrial depolarization and bioenergetic failure[63], contributing to the pathogenesis of neurodegenerative diseases, including HD. The calpain-mediated cleavage of p35 to p25 activates CDK5, which then binds to and phosphorylates Drp1, leading to Drp1 recruitment to the mitochondria and resultant fragmentation[64]. Activated calpain may also directly cleave Drp1, possibly contributing to mitochondrial fragmentation and neuronal damage in AD models[65]. The inhibition of calpain by CAST appears to preserve mitochondrial morphology by reducing the mitochondrial fission proteins, Drp1 and Fis1, and protecting neurons against excito-toxic cell death[61]. Additionally, CAST also prevents the mito-chondrial translocation of Drp1 by inhibiting calpain-mediated calcineurin activation[66]. A recent study reported that CAST was also distributed on the mitochondria–ER-associated membrane and was transported along neuronal axons by the mitochondrial fusion-related protein, mitofusin 2, to prevent synaptic elimination during ALS[67]. Thus, the protection of CAST from neuronal death is involved in the inhibition of mitochondrial impairment by balancing mitochondrial fission and fusion processes. In this study, we further showed that CHIR99021 treatment suppressed Drp1 hyperactivation and resultant mitochondrial fragmentation. Interestingly, these events occurred in a CAST-dependent manner; CAST loss completely abolished CHIR99021-mediated inhibition of Drp1 activity and improved mitochondrial morphology. Treatment with a CAST-derived peptide that inhibits calpain activation consistently reduced mitochondrial fragmentation, mitochondrial depolarization, and cell death in our HD models. Therefore, CAST may be a key molecule upstream of Drp1 that mediates mitochondrial protection and neuroprotection induced by CHIR99021 in HD models.

In summary, we used a chemical screen to identify CHIR99021 as a potent mitochondrial enhancer that suppressed HD-associated neuropathology. Furthermore, we identified a CAST–calpain–Drp1 signaling pathway regulated by CHIR99021. The protective effects of calpain inhibition via the genetic enhancement of CAST have been observed in several other neurodegenerative diseases, such as AD, PD, and ALS. Further investigations on how the CAST–calpain–Drp1 cell death pathway is controlled and how the biological optimization of CHIR99021 may accelerate the discovery of therapeutic approaches to treat multiple neurodegenerative diseases are warranted.

## Methods

**Reagents and antibodies**. Protein phosphatase inhibitor cocktail (P5726), protease inhibitor cocktails (8340), and 3-NP (N5636) were purchased from Milli-poreSigma (Burlington, MA, USA). CHIR99021 (S2924) was from Selleckchem (Houston, TX, USA). MG132 (13697), lactacystin (70980), epoxomicin (10007806), BFA (11038), tunicamycin (11445), and thapsigargin (10522) were purchased from Cayman Chemical (Ann Arbor, MI, USA). *N*-acetyl-calpastatin (184-210) (J66680MCR) was purchased from Alfa Aesar (Haverhill, MA, USA). The antibodies used in the study are listed below: Anti-DARPP-32 (ab40801, Abcam, Cambridge, UK, 1:3000), anti-BDNF (ab108319, Abcam, 1:1000), anti-PGC1α (ab54481, Abcam, 1:1000), anti-VDAC1 (14734, Abcam, 1:2000), anti-CDK5 (ab40773, Abcam, 1:1000), anti-MCL1 (16225-1-AP, Proteintech, Rosemont, IL, USA, 1:1000), anti-Mff (17090-1-AP, Proteintech, 1:2000), anti-p62 (18420-1-AP, Proteintech, 1:5000), anti-Fis1 (10956-1-AP, Proteintech, 1:1000), anti-MIEF (20164-1-AP, Proteintech, 1:1000), anti-TFEB (13372-1-AP), anti-ATPB (17247-1-AP, Proteintech, 1:3000), anti-MFN1 (H00055669-M04, Abnova, Taipei, Taiwan,

1:2000), anti-GRP78 (SPA-827, Stressgen Enzo Life Sciences, Farmingdale, NY, USA, 1:1000), anti-GSK3 (05-412, MilliporeSigma, 1:1000), anti-huntingtin protein (MAB5374, clone EM48, MilliporeSigma, 1:1000), anti-β-actin (A1978, MilliporeSigma, 1:10,000), anti-DLP1 (611113, BD Bioscience, Franklin Lakes, NJ, USA, 1:2000), anti-GSK3 pY216/279 (612312, BD Bioscience, 1:1000), anti-SOD2 (611580, BD Bioscience, 1:1000), anti-OPA1 (612607, BD Bioscience, 1:2000) anti-PSD95 (2507, Cell Signaling, Danvers, MA, USA, 1:5000), anti-Drp1-pS616 (3455S, Cell Signaling, 1:1000), anti-p35/25 (2680, Cell Signaling, 1:1000), anti-LC3 (2775 S, Cell Signaling, 1:1000), anti-CAST (4146, Cell Signaling, 1:2000), anti-calpain I (2556, Cell Signaling, 1:1000), anti-ubiquitin (43124S, Cell Signaling, 1:1000), anti-spectrin-α II (sc-48382, Santa Cruz Biotechnology, Dallas, TX, USA, 1:500), anti-CHOP (sc575, Santa Cruz Biotechnology, 1:2000), and anti-enolase (sc-15343, Santa Cruz Biotechnology, 1:2000), horseradish peroxidase (HRP)-conjugated anti-rabbit or anti-mouse IgG (31430/31460, ThermoFisher Scientific, 1:5000).

**Cell culture**. Striatal neuron-like WT HdhQ7 and HD mutant HdhQ111 cells were obtained from the Cure Huntington's Disease Initiative (CHDI) Foundation and were cultured in Dulbecco's modified Eagle's medium (DMEM) supplemented with 10% fetal bovine serum (FBS), 100 mg/mL penicillin, 100 mg/mL streptomycin, and 400 µg/mL G418. Cells were grown at 33 °C in a 5% $CO_2$ environment and were used within 14 passages for studies.

HEK293 (ATCC, CRL-1573) and Neuro2a (ATCC, CCL-131) cells were maintained in DMEM supplemented with 10% FBS and 1% penicillin/streptomycin (v/v) at 37 °C in a 5% $CO_2$ environment and were used within 10 passages for studies. Stable Neuro2a cells were maintained as described plus 1 µg/mL puromycin (Corning, NY, USA).

HD patient fibroblasts (4208, 4693, 21756, 3621, 5539; Coriell Institute, Camden, NJ, USA) and normal fibroblasts (nHDF, HDF, Huf1; Coriell Institute) were maintained in Minimum Essential Medium supplemented with 15% (v/v) FBS and 1% (v/v) penicillin/streptomycin at 37 °C in 5% $CO_2$.

iPS cells from NS and patients with HD were differentiated into neurons using a protocol from our previous study[14,18]. Briefly, iPS cells were plated onto 6-well plates pre-coated with 2.5% matrigel (Corning) and allowed reach 90% confluence in feeder-free medium (mTeSR™ Plus, 100-0276, STEMCELL Technologies, Vancouver, Canada). For the first 10 days, cells were treated with SB431542 (10 mM; Tocris Bioscience, Bristol, UK) and Noggin (100 ng/mL; R&D systems, Minneapolis, MI, USA) in neural medium containing neurobasal™ A medium (ThermoFisher Scientific, Waltham, MA) and DMEM/F12 GlutaMAX™ (ThermoFisher Scientific) (1:1), B27 supplement minus vitamin A (50×, Invitrogen, Carlsbad, CA, USA), N2 supplement (100×, Invitrogen), GlutaMax (100×, Invitrogen), human recombinant fibroblast growth factor—basic (20 ng/mL, PEPROTECH, Rocky Hill, NJ, USA) and human recombinant epidermal growth factor (20 ng/mL, MilliporeSigma), 100 units/mL penicillin, and 100 µg/mL streptomycin. For the next 10 days, cells were treated with human recombinant Sonic hedgehog (200 ng/mL, PEPROTECH), human recombinant DKK1 (100 ng/mL, PEPROTECH) and human recombinant BDNF (20 ng/mL, PEPROTECH), and 10 mM Y27632 (MilliporeSigma) in neuronal differentiation medium containing neurobasal™ A medium and DMEM/F12 Glutamax™ medium (1:3), B27, N2, GlutaMax, and 50 units/mL penicillin and 50 µg/mL streptomycin. Cells were then switched to treatment with BDNF (20 ng/mL), ascorbic acid (200 mM, MilliporeSigma), dibutyryl cAMP (0.5 mM, MilliporeSigma), and Y27632 (10 µM) in neuronal differentiation medium. Twenty days after differentiation initiation, neurons (approximately 5000 cells) were plated onto 12-mm poly-D-lysine (MilliporeSigma)/laminine (ThermoFisher Scientific)-coated coverslips and grown in 24-well plates in neuronal differentiation medium. After a 5-day treatment with CHIR99021 (Selleckchem, Houston, TX, USA) at 1 µM, cells were fixed in 4% paraformaldehyde and subjected to immunostaining. To measure cell survival, the same number of neurons were plated in 96-well plates. After a 4-day treatment with 1 µM CHIR99021, BDNF was withdrawn from the medium and cell viability was measured after 12 h using the 3-(4,5-dimethylthiazol-2-yl)-2,5-diphenyl-2H-tetrazolium bromide assay (Roche, Indianapolis, IA, USA) following the manufacturer's instructions.

**High-throughput screening platform**. HdhQ111 striatal cells were grown and expanded in culture flasks before harvesting for plating. Cells were dispensed into fibronectin-coated 384-well plates (Corning) using a Biotek EL406 Microplate Washer Dispenser (Biotek, Winooski, VT, USA) to a final density of 1000 cells/well. After overnight incubation, small molecules were added to assay plates at 10 mM in DMSO via a 50 nL slotted pin tool attached to a Janus automated workstation (Perkin Elmer, Waltham, MA, USA), resulting in a final screening concentration of 10 µM.

For the primary screen, after a 2-day treatment with small molecules, cell medium was replaced with fresh medium containing TMRM dye (0.25 µM) (T668, Thermo Fisher Scientific), a MMP indicator, and Hoechst 33342 (5 µg/mL) (Invitrogen) for 20 min at 33 °C. Cells were washed twice in phosphate-buffered saline (PBS) pH 7.4 and imaged using an Operetta High Content Imaging and Analysis system (Perkin Elmer). Four fields were captured from each well resulting in an average of 700 cells scored/imaged/well. Image analysis (Acapella software (Perkin Elmer)) was initiated by identifying intact nuclei stained by Hoechst by

nuclei image area (>300 µm$^2$). Each nucleus region mask was then expanded by 50% and cross-referenced with TMRM staining, and from this, the TMRM fluorescence density was calculated. Molecules inducing more than a 50% increase in TMRM density relative to DMSO control wells were selected, thus hits were assigned based on the largest fold increase in TMRM fluorescent density percentages relative to DMSO controls in the same plate.

For the secondary screen, cells were incubated for 2 days with leading TMRM screen hits at concentrations spanning 0.078–10 µM. Then the culture medium was replaced with fresh medium without FBS for 16 h, cells were stained with Hoechst dye, and images were acquired using the Operetta high-content imaging system (Perkin Elmer). Pyknotic nuclei were scored and nuclei percentages were determined using the Acapella image analysis software (Perkin Elmer) to provide an assessment of HdhQ111 cell death. Hits were assigned based on the largest fold decrease in pyknotic nuclei percentages relative to DMSO controls in the same plate.

**MMP measurements**. Cells cultured on coverslips were washed in PBS (pH 7.4) and incubated with 0.25 µM TMRM and 5 µg/mL Hoechst for 20 min at 33 °C for mouse striatal cells and 37 °C for other cells. Images were visualized by confocal microscopy (Olympus, Tokyo, Japan; Fluoview FV100) and image quantification was performed using the ImageJ software. At least 100 cells/group were counted for analysis. TMRM fluorescence density was normalized to the total number of cells.

**Measurement of mitochondrial respiratory capacity**. Mouse striatal HdhQ7/Q111 cells were seeded in XFp 8-well miniplates (103025-100, Agilent, Santa Clara, CA, USA) at 3000 cells/well in 100 µL growth medium. Two days after treatment with 3 µM CHIR99021, mitochondrial respiration activity in intact cells was analyzed using a Seahorse Bioscience XFp Extracellular Flux analyzer (Agilent). Briefly, 1 h prior to measuring oxygen consumption, cell culture media was replaced with XF assay medium and maintained in a non-$CO_2$ incubator for 1 h at 33 °C. Sensor cartridges were placed in the XFp analyzer according to the manufacturer's instructions from the Mito Stress Test Kit (Agilent, 103010-100). Mitochondrial function was determined by the sequential injection of oligomycin A (1 µM), FCCP (1 µM), and rotenone/antimycin A (0.5 µM). The total protein content in each well was determined after respiration measurement, and all results were normalized to the total protein content.

**Mitochondrial ROS measurement**. Mouse striatal cells cultured on coverslips were washed in PBS (pH 7.4) and incubated with 5 µM MitoSOX™ Red (Invitrogen, M36008), a mitochondrial superoxide indicator, and 5 µg/mL Hoechst for 10 min at 33 °C. Images were visualized by confocal microscopy (Olympus) and image quantification was performed using the ImageJ software. At least 100 cells/group were counted for analysis. MitoSOX fluorescence density was normalized to the total number of cells.

**Cell death measurement**. Mouse striatal HdhQ7/HdhQ111 cells were treated with 3 µM CHIR99021 for 2 days and then serum-starved for 16 h. Cell death was determined by measuring LDH release into the medium using an LDH-cytotoxicity assay kit II (Roche) following the manufacturer's instructions. Relative LDH release was calculated as: $LDH_{medium}/(LDH_{medium} + LDH_{cell})$ and then normalized to HdhQ7 cells data.

**In vitro calpain activity**. Endogenous calpain enzymatic activity in HdhQ7/Q111 cells, and R6/2 and YAC128 mouse striatum, and recombinant calpain substrate cleavage in the absence/presence of CHIR99021 was measured using the calpain activity fluorometric assay kit (BioVision, Milpitas, CA, USA) following the manufacturer's instructions.

**In vivo and in vitro proteasome activity**. For the in vivo measurement of proteasome activity, HdhQ7 and HdhQ111 cells were treated with CHIR99021 for 48 h. For in vitro measurements, the total proteasome fraction was extracted from cells followed by incubation with CHIR99021 for 30 min. Proteasome activity was assessed using the proteasome activity fluorometric assay kit (BioVision) following the manufacturer's instructions.

**Real-time PCR**. Total RNA was isolated using the RNeasy Mini Kit (QIAGEN, Hilden, Germany), and 0.5–1 µg of total RNA was used to synthesize cDNA using the QuantiTect Reverse Transcription Kit (QIAGEN). qRT-PCR was performed using QuantiTect SYBR Green (QIAGEN) and analyzed with the StepOnePlus Real-Time PCR System (ThermoFisher Scientific). Three replicates were performed for each biological sample, and the expression values of each replicate were normalized against GAPDH cDNA using the $2^{-\Delta\Delta CT}$ method. The primers used are provided in Supplementary Table 1.

**Animal models of HD**. All animal studies were conducted in accordance with protocols approved by the Institutional Animal Care and Use Committee of Case Western Reserve University and were performed based on the NIH Guide for the

Care and Use of Laboratory Animals. Sufficient procedures were employed for reducing pain or discomfort of mice during the experiments.

Male R6/2 mice and their WT littermates (4 weeks old) were purchased from The Jackson Laboratory (Bar Harbor, ME, USA; B6CBA-TgN [HD exon 1]); JAX stock number: 006494). R6/2 mice (C57BL/6 and CBA genetic background) are transgenic for the 5' end of the human *HD* gene carrying 100–150 glutamine (CAG) repeats. Male R6/2 mice at 6–12 weeks were used in the study. YAC128 (FVB-Tg (YAC128) 53Hay/J, JAX stock number: 004938) breeders (FVB/N genetic background) were also purchased from The Jackson Laboratory. YAC128 mice contained a full-length human *HD* gene modified with a 128 CAG repeat expansion in exon 1. The mice were mated, bred, and genotyped in the animal facility of Case Western Reserve University. Male YAC128 mice at 9–12 months were used in the study. All mice were maintained with a 12-h light/dark cycle (on at 06:00 hours and off at 18:00 hours).

**CHIR99021 treatment in HD mice**. All randomization and compound treatments were prepared by a researcher not associated with behavioral and neuropathological analyses. For R6/2 mice, male hemizygous HD R6/2 mice and their age-matched WT littermates were given an i.p. injection of CHIR99021 (10 mg/kg dissolved in 15% captisol (CAPTISOL, Kansas, USA)) or vehicle (15% calptisol) once a day for 5 days/week starting at 6 weeks old. At 12 weeks, the treatment was terminated.

For YAC128 mice, male HD YAC128 and their age-matched WT littermates were given i.p. injections of CHIR99021 (10 mg/kg dissolved in 15% captisol) or vehicle (15% calptisol) once every other day starting at 9 months old and terminating at 12 months old.

**Behavioral analysis in HD mice**. All behavioral analyses were conducted by a researcher blinded to genotypes and treatment groups, as we previously described[18]. Gross locomotor activity was assessed in R6/2 mice and age-matched WT littermates at 6, 9, and 12 weeks old. In an open-field activity chamber (Omnitech Electronics Inc., Columbus, OH, USA), mice were placed in the center of the chamber and allowed to explore while being tracked by an automated beam system (Vertax, Omnitech Electronics Inc.). Horizontal and vertical distances and rearing activities were all recorded. Because R6/2 mice were sensitive to changes in the environment and handling, we only conducted a 1 h locomotor activity analysis for these mice and WT littermates. Hindlimb clasping was assessed by a tail suspension test once a week from 9 to 12 weeks old. Briefly, mice were suspended for 20 s and hindlimb latency or four paw clasping was recorded using a scoring system: clasping over 10 s, score 3; 5–10 s, score 2; 0–5 s, score 1; and 0 s, score 0. Body weights and survival rates of R6/2 mice and WT littermates were recorded throughout the study period.

Locomotor activity was assessed in YAC128 mice and age-matched WT littermates at 9–12 months old. Motor coordination was assessed in YAC128 mice and age-matched WT littermates by measuring latency on a rotarod in an accelerated program for 300 s every month starting from 9 to 12 months old. Body weights of YAC128 mice and WT littermates were recorded throughout the study period.

**Constructs and transfection**. Myc-tagged full-length Htt with 23Q (CH00038), 73Q (CH00039), or 145Q (CH00040) repeat constructs were obtained from the CHDI Foundation. Cells were transfected using TransIT-2020 (Mirus Bio, LLC, Madison, WI, USA) transfection reagent following the manufacturer's instructions.

**RNA interference**. GSK3 siRNA (1320001) and control siRNA were purchased from Fisher Scientific (Hampton, NH, USA). CDK5 siRNA (L-040544-00-0005) was purchased from Dharmacon (Lafayette, Co, USA). CAST shRNA (TRCN0000080114), Calpain I shRNA (TRCN0000030664), and control shRNA were purchased from MilliporeSigma. For GSK3 and CDK5 siRNA delivery, cells were transfected with TransIT-TKO transfection reagent (Mirus Bio) and maintained for 3 days before studies. Cells were infected with lentiviral particles containing CAST shRNA or Calpain I shRNA for 2 days and selected using puromycin (2 μg/mL) (Corning) to generate a stable CAST or Calpain I knock-down line.

To knock down CAST in neurons differentiated from human iPSCs, lentiviral particles containing control or CAST shRNA (Origene, Rockville, MD, USA) were added into medium (MOI 12:1) with polybrene (4 μg/mL) (MilliporeSigma). Medium was changed after 12 h infection. Neurons were infected again 24 h after the first infection and subjected for studies after 48 h.

**GSK3β KO by CRISPR-Cas9**. To silence *GSK3B* in Neuro2a cells, a control and *GSK3B* synthetic guide RNA (sgRNA) CRISPR lenti-vector set (K5030401) was purchased from Applied Biological Materials (Richmond, BC, Canada). Neuro2a cells were transfected as described with either control or *GSK3B* sgRNA. Stable Neuro2a cells with *GSK3B* KO were selected using puromycin (Corning) at 4 μg/mL and maintained at 1 μg/mL.

**Sodium dodecyl sulfate-polyacrylamide electrophoresis (SDS-PAGE) and western blotting**. Protein concentrations were determined by Bradford assay (Bio-

Rad Laboratories, Hercules, CA, USA). Protein (15–25 μg) was resuspended in 5× Laemmli buffer, boiled at 100 °C for 5 min, and subjected to SDS-PAGE. Separated proteins were then transferred to nitrocellulose membranes (Bio-Rad Laboratories) and blocked for 1 h in 5% non-fat milk in Tris-buffered saline containing 0.1% Tween 20 (TBST). Membranes were then probed overnight with the primary antibodies. After washing three times in TBST, membranes were incubated for 1 h at room temperature with secondary anti-rabbit or anti-mouse IgG (31430/31460, ThermoFisher Scientific, 1:5000), followed by visualization with enhanced chemiluminescence. Representative blots have been cropped for presentation.

**Dot blot analysis**. The triton X-100 insoluble fractions isolated from the striatum of R6/2 and YAC128 mice were washed with total lysis buffer three times and solubilized in 1% SDS using probe sonication (35% amplitude, 15 s) (Qsonica, Newtown, CT, USA). One microliter of samples was spotted onto the nitrocellulose membrane (Bio-Rad). After it dried, the membrane was blocked for 1 h in 5% non-fat milk in TBST. Membranes were then probed overnight with anti-huntingtin protein antibody (MAB5374, clone EM48, MilliporeSigma). After washing three times in TBST, membranes were incubated for 1 h at room temperature with poly (HRP)-conjugated anti-mouse IgG (31460, ThermoFisher Scientific, 1:5000), followed by visualization with enhanced chemiluminescence. Representative blots have been cropped for presentation.

**Immunohistochemistry**. Mice were deeply anesthetized and trans-cardially perfused with 4% paraformaldehyde in PBS. Brains were processed for paraffin embedding. Brain sections (10 μm, coronal) were hydrated and de-paraffinized by sequential incubation in xylene (three times), 100% ethanol (twice), 95% ethanol, 70% ethanol, 50% ethanol, and water (twice). After antigen retrieval in 0.01 M sodium citrate buffer plus 0.05% Tween 20 (pH 6.0), slides were incubated with 3% hydrogen peroxide ($H_2O_2$) in methanol to quench endogenous peroxidase. They were then treated with 5% normal goat serum (Invitrogen) in TBST for 1 h at room temperature. Sections were then incubated with anti-DARPP-32 (1710-1, Epitomics, Burlingame, CA, USA; 1:500) and anti-huntingtin protein (MAB5374, clone EM48, MilliporeSigma, 1:500) in a humidified chamber overnight at 4 °C. The next day, slides were incubated with biotin-conjugated secondary antibody (goat anti-mouse/rabbit) and streptavidin-conjugated HRP using an immunohistochemistry select HRP/DAB kit (MilliporeSigma, DAB150) and DAB solution following the manufacturer's instructions. Slides were mounted after sequential dehydration in water, 50% ethanol, 70% ethanol, 95% ethanol, 100% ethanol, and xylene. Images were captured using a digital microscope (VHX-7000, Keyence, Osaka, Japan). Quantitation of DARPP-32 and mutant huntingtin (mtHtt) aggregates/100 $μm^2$ immunostained area was conducted using the ImageJ software. The same image exposure times and threshold settings were used for all the group sections. A researcher blinded to experimental groups conducted quantification analyses.

**Immunocytochemistry**. Cells cultured on coverslips were washed in PBS (pH 7.4), fixed in 4% paraformaldehyde, and permeabilized in 0.1% Triton X-100. After incubation with 2% normal goat serum, fixed cells were incubated overnight at 4 °C with the following primary antibodies: anti-Tau (T1308-1, rPeptide, Athens, Georgia, USA 1:200), anti-DARPP-32 (ab40801, Abcam, 1:500), anti-MAP2 (4542, Cell Signaling, 1:500), anti-GAD67 (MAB5406, MilliporeSigma, 1:300), anti-Tubulin β 3 (Tuj1) (801201, BioLegend, San Diego, CA, USA, 1:500), and anti-Tom20 (11802-1-AP, Proteintech, 1:1000). Cells were washed in PBS (pH 7.4) and incubated with Alexa Fluor goat anti-mouse/rabbit 568 or 488 secondary antibodies (ThermoFisher Scientific, 1:1000), followed by incubation with Hoechst dye (1:10,000). Coverslips were mounted and slides were imaged by confocal microscopy (Olympus). $MAP2^+$ neurite length, $Tau^+/Tuj1^+$ axonal length, and mitochondrial morphology were quantified using the ImageJ software.

**Preparation of total lysates**. Cells were washed in cold PBS (pH 7.4) and incubated on ice for 30 min in total lysis buffer (50 mM Tris-HCl, pH 7.5, 150 mM NaCl, 1% Triton X-100, protease inhibitors cocktail, and phosphatase inhibitors cocktail (MilliporeSigma)). Mouse brains were minced and homogenized in lysis buffer and placed on ice for 30 min. Cells or tissues were centrifuged at 12,000 × g for 10 min at 4 °C to generate total lysate supernatants.

**Immunoprecipitation**. Cells were harvested and lysed in total lysis buffer for 30 min on ice and centrifuged at 12,000 × g for 10 min at 4 °C. The supernatants were incubated with anti-CAST antibody overnight at 4 °C, followed by incubation with protein A/G beads (sc-2003, Santa Cruz Biotechnology) for 2 h at 4 °C. The immunoprecipitates were washed with lysis buffer three times for a total of 30 min and then subject to western blotting.

**Isolation of subcellular fractions**. Cells were washed in cold PBS and incubated on ice for 30 min in subcellular lysis buffer (250 mM sucrose, 20 mM HEPES-NaOH, pH 7.5, 10 mM KCl, 1.5 mM $MgCl_2$, 1 mM EDTA, protease and phosphatase inhibitor cocktails (MilliporeSigma)). Mouse brains were minced and homogenized in lysis buffer and placed on ice for 30 min. Cells or tissues were

disrupted 20 times by repeated aspiration through a 25-gauge needle, followed by a 30-gauge needle. Homogenates were centrifuged at $800 \times g$ for 10 min at 4 °C and resulting supernatants further centrifuged at $10,000 \times g$ for 20 min at 4 °C. Pellets were washed in lysis buffer and re-centrifuged at $10,000 \times g$ for 20 min at 4 °C. Final pellets were suspended in lysis buffer containing 1% Triton X-100 and were assigned as mitochondrial-rich lysate fractions. Supernatants were designated as cytosolic fractions.

**Label-free proteomics.** HhdQ111 and Q7 cells were harvested and lysed with total lysis buffer. Following lysis, the samples were processed using a filter-aided sample preparation clean-up protocol[68] with Amicon Ultra molecular weight cut-off 3K filters (Millipore, Billerica, MA, USA). Samples were reduced and alkylated on filters using 10 mM dithiothreitol (Acros, Fair Lawn, NJ, USA) and 25 mM iodoacetamide (Acros), respectively, and then concentrated to a final volume of 40 μL in 8 M urea. Protein concentrations were measured using the Bradford method according to the manufacturer's instructions (Bio-Rad).

Following reduction and alkylation, total protein (10 μg) was subjected to enzymatic digestion. The urea concentration was adjusted to 4 M using 50 mM Tris (pH 8) and proteins were digested using mass spectrometry-grade lysyl endopeptidase (Wako Chemicals, Richmond, VA, USA) at an enzyme/substrate ratio of 1:40 for 2 h at 37 °C. Then the urea concentration was further adjusted to 2 M using 50 mM Tris (pH 8) and lysyl peptides were digested overnight 37 °C in sequencing-grade trypsin (Promega, Madison, WI, USA) at an enzyme/substrate ratio of 1:40. Finally, samples were diluted in 0.1% formic acid (Thermo Scientific, Rockford, IL, USA) before liquid chromatography-tandem mass spectrometry (LC-MS/MS) analysis.

The 300 ng of lysed protein from each of group were loaded onto a column in a 3 μL injection volume with blanks in between for a total of four LC/MS/MS runs. Data were acquired with an Orbitrap Velos Elite mass spectrometer (Thermo Electron, San Jose, CA, USA) equipped with a Waters nanoACQUITY LC system (Waters, Taunton, MA, USA). Peptides were desalted on a trap column (180 μm × 20 mm, packed with C18 Symmetry, 5 μm, 100 Å, Waters) and subsequently resolved on a reversed-phase column (75 μm × 250 mm nano column, packed with C18 BEH130, 1.7 μm, 130 Å (Waters)). LC was conducted at an ambient temperature at a flow rate of 300 nL/min using a gradient mixture of 0.1% formic acid in water (solvent A) and 0.1% formic acid in acetonitrile (solvent B). The gradient ranged from 4 to 44% solvent B over 210 min. Peptides eluting from the capillary tip were introduced into the nanospray mode at a capillary voltage of 2.4 kV. A full scan was obtained for eluted peptides in the range of 380–1800 atomic mass units, followed by 25 data-dependent MS/MS scans. MS/MS spectra were generated by collision-induced dissociation of peptide ions at a normalized collision energy of 35% to generate a series of b- and y-ions as major fragments. In addition, a 1 h wash was included between samples. The proteins were identified with Mascot (Matrix Sciences, London, UK). Key search parameters were Trypsin for enzyme; a maximum of 1 missed cleavage; peptide charge states of +2 to +3; peptide tolerance of 10 p.p.m.; and MS/MS tolerance of 0.8 Da. Oxidation of methoinine was a variable modification. Identifications were merged and the spectra were summed and then quantified using spectral counting (selecting normalized total spectra) in Scaffold version 4.4.0 (Proteome Software, Inc., Portland, OR). The protein probability was determined using a protein threshold of 99%, two peptides, and a peptide threshold of 95%[69,70]. A false discovery rate of 0.86% was calculated using ProteinProphet algorithm for the 56,902 spectra examined[70]. Both MASCOT and Scaffold strategies utilized the Uniprot human database from June 2016 (20,199 sequences).

**Quantification and statistical analysis.** Sample sizes were determined by a power analysis based on pilot data collected in our laboratory or from published studies. For animal studies, we used $n = 10–15$ mice/group for behavioral tests, $n = 3–6$ mice/group for biochemical analyses, and $n = 3–10$ mice/group for pathology studies. In cell culture studies, each experiment was independently conducted at least three times. For animal studies, we ensured randomization and blinded evaluations. For imaging studies, a blinded observer performed quantification analyses. No samples or animals were excluded from our analysis.

Data were analyzed using GraphPad Prism 9 (GraphPad Software, San Diego, CA, USA). Unpaired Student's $t$ test was used for comparisons between two groups. Comparisons between three or more independent groups were performed using one-way ANOVA, followed by Tukey's post hoc test. Comparisons of the effect of independent variables on a response variable were performed using two-way ANOVA. Survival rate was analyzed by Log-rank (Mantel–Cox) test. All values are reported as mean ± standard error of the mean (SEM). Data are representative of at least three independent experiments. Statistical parameters are presented in each figure legend. We considered $p < 0.05$ as statistically significant.

**Reporting summary.** Further information on research design is available in the Nature Research Reporting Summary linked to this article.

## Data availability

Data supporting the findings of this study are provided within the paper and its Supplementary Information. The proteomic database was submitted to figshare (https://

figshare.com/) with https://doi.org/10.6084/m9.figshare.14850267.v1. Source data are provided with this paper.

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

# ARTICLE

25. Gauthier, L. R. et al. Huntingtin controls neurotrophic support and survival of neurons by enhancing BDNF vesicular transport along microtubules. *Cell* **118**, 127–138 (2004).

26. Slow, E. J. et al. Selective striatal neuronal loss in a YAC128 mouse model of Huntington disease. *Hum. Mol. Genet.* **12**, 1555–1567 (2003).

27. Wang, H., Chen, X., Li, Y., Tang, T. S. & Bezprozvanny, I. Tetrabenazine is neuroprotective in Huntington's disease mice. *Mol. Neurodegener.* **5**, 18 (2010).

28. Ross, C. A. & Tabrizi, S. J. Huntington's disease: from molecular pathogenesis to clinical treatment. *Lancet Neurol.* **10**, 83–98 (2011).

29. Cline, G. W. et al. Effects of a novel glycogen synthase kinase-3 inhibitor on insulin-stimulated glucose metabolism in Zucker diabetic fatty (fa/fa) rats. *Diabetes* **51**, 2903–2910 (2002).

30. Ying, Q. L. et al. The ground state of embryonic stem cell self-renewal. *Nature* **453**, 519–523 (2008).

31. Fernandez-Nogales, M. et al. Decreased glycogen synthase kinase-3 levels and activity contribute to Huntington's disease. *Hum. Mol. Genet.* **24**, 5040–5052 (2015).

32. Wood, N. I. & Morton, A. J. Chronic lithium chloride treatment has variable effects on motor behaviour and survival of mice transgenic for the Huntington's disease mutation. *Brain Res. Bull.* **61**, 375–383 (2003).

33. Joshi, A. U., Ebert, A. E., Haileselassie, B. & Mochly-Rosen, D. Drp1/Fis1-mediated mitochondrial fragmentation leads to lysosomal dysfunction in cardiac models of Huntington's disease. *J. Mol. Cell. Cardiol.* **127**, 125–133 (2019).

34. Wagner, F. F. et al. Exploiting an Asp-Glu "switch" in glycogen synthase kinase 3 to design paralog-selective inhibitors for use in acute myeloid leukemia. *Sci. Transl. Med.* **10**, eaam8460 (2018).

35. Hanna, R. A., Campbell, R. L. & Davies, P. L. Calcium-bound structure of calpain and its mechanism of inhibition by calpastatin. *Nature* **456**, 409–412 (2008).

36. Baudry, M. & Bi, X. Calpain-1 and Calpain-2: the yin and yang of synaptic plasticity and neurodegeneration. *Trends Neurosci.* **39**, 235–245 (2016).

37. Menzies, F. M. et al. Calpain inhibition mediates autophagy-dependent protection against polyglutamine toxicity. *Cell Death Differ.* **22**, 433–444 (2015).

38. Brouillet, E. et al. Chronic mitochondrial energy impairment produces selective striatal degeneration and abnormal choreiform movements in primates. *Proc. Natl Acad. Sci. USA* **92**, 7105–7109 (1995).

39. Nath, R. et al. Non-erythroid alpha-spectrin breakdown by calpain and interleukin 1 beta-converting-enzyme-like protease(s) in apoptotic cells: contributory roles of both protease families in neuronal apoptosis. *Biochem. J.* **319**, 683–690 (1996).

40. Lee, M. S. et al. Neurotoxicity induces cleavage of p35 to p25 by calpain. *Nature* **405**, 360–364 (2000).

41. Paoletti, P. et al. Dopaminergic and glutamatergic signaling crosstalk in Huntington's disease neurodegeneration: the role of p25/cyclin-dependent kinase 5. *J. Neurosci.* **28**, 10090–10101 (2008).

42. Wagner, F. F. et al. Inhibitors of glycogen synthase kinase 3 with exquisite kinome-wide selectivity and their functional effects. *ACS Chem. Biol.* **11**, 1952–1963 (2016).

43. Maki, M. et al. Inhibition of calpain by a synthetic oligopeptide corresponding to an exon of the human calpastatin gene. *J. Biol. Chem.* **264**, 18866–18869 (1989).

44. Guo, X. & Qi, X. VCP cooperates with UBXD1 to degrade mitochondrial outer membrane protein MCL1 in model of Huntington's disease. *Biochim. Biophys. Acta Mol. Basis Dis.* **1863**, 552–559 (2017).

45. Chang, C. R. & Blackstone, C. Dynamic regulation of mitochondrial fission through modification of the dynamin-related protein Drp1. *Ann. NY Acad. Sci.* **1201**, 34–39 (2010).

46. Meijer, L., Flajolet, M. & Greengard, P. Pharmacological inhibitors of glycogen synthase kinase 3. *Trends Pharmacol. Sci.* **25**, 471–480 (2004).

47. An, W. F. et al. Discovery of Potent and Highly Selective Inhibitors of GSK3b. In: *Probe Reports from the NIH Molecular Libraries Program* (NCBI, 2012).

48. Ring, D. B. et al. Selective glycogen synthase kinase 3 inhibitors potentiate insulin activation of glucose transport and utilization in vitro and in vivo. *Diabetes* **52**, 588–595 (2003).

49. Bar-Nur, O. et al. Lineage conversion induced by pluripotency factors involves transient passage through an iPSC stage. *Nat. Biotechnol.* **33**, 761–768 (2015).

50. Ma, Y. et al. CHIR-99021 regulates mitochondrial remodelling via beta-catenin signalling and miRNA expression during endodermal differentiation. *J. Cell Sci.* **132**, jcs229948 (2019).

51. Moldoveanu, T. et al. A Ca(2+) switch aligns the active site of calpain. *Cell* **108**, 649–660 (2002).

52. Gafni, J. & Ellerby, L. M. Calpain activation in Huntington's disease. *J. Neurosci.* **22**, 4842–4849 (2002).

53. Cho, K. et al. Calpain-mediated cleavage of DARPP-32 in Alzheimer's disease. *Aging cell* **14**, 878–886 (2015).

54. Yildiz-Unal, A., Korulu, S. & Karabay, A. Neuroprotective strategies against calpain-mediated neurodegeneration. *Neuropsychiatr. Dis. Treat.* **11**, 297–310 (2015).

55. Weber, J. J. et al. Calpastatin ablation aggravates the molecular phenotype in cell and animal models of Huntington disease. *Neuropharmacology* **133**, 94–106 (2018).

56. Doumit, M. E. & Koohmaraie, M. Immunoblot analysis of calpastatin degradation: evidence for cleavage by calpain in postmortem muscle. *J. Anim. Sci.* **77**, 1467–1473 (1999).

57. Suwanjang, W., Phansuwan-Pujito, P., Govitrapong, P. & Chetsawang, B. Calpastatin reduces calpain and caspase activation in methamphetamine-induced toxicity in human neuroblastoma SH-SY5Y cultured cells. *Neurosci. Lett.* **526**, 49–53 (2012).

58. Ni, R. et al. Mitochondrial calpain-1 disrupts ATP synthase and induces superoxide generation in type 1 diabetic hearts: a novel mechanism contributing to diabetic cardiomyopathy. *Diabetes* **65**, 255–268 (2016).

59. Cao, G. et al. Critical role of calpain I in mitochondrial release of apoptosis-inducing factor in ischemic neuronal injury. *J. Neurosci.* **27**, 9278–9293 (2007).

60. Smith, M. A. & Schnellmann, R. G. Calpains, mitochondria, and apoptosis. *Cardiovasc. Res.* **96**, 32–37 (2012).

61. Tangmansakulchai, K. et al. Calpastatin overexpression reduces oxidative stress-induced mitochondrial impairment and cell death in human neuroblastoma SH-SY5Y cells by decreasing calpain and calcineurin activation, induction of mitochondrial fission and destruction of mitochondrial fusion. *Mitochondrion* **30**, 151–161 (2016).

62. Detmer, S. A. & Chan, D. C. Functions and dysfunctions of mitochondrial dynamics. *Nat. Rev. Mol. Cell Biol.* **8**, 870–879 (2007).

63. Liesa, M. & Shirihai, O. S. Mitochondrial dynamics in the regulation of nutrient utilization and energy expenditure. *Cell Metab.* **17**, 491–506 (2013).

64. Meuer, K. et al. Cyclin-dependent kinase 5 is an upstream regulator of mitochondrial fission during neuronal apoptosis. *Cell Death Differ.* **14**, 651–661 (2007).

65. Jiang, S., Shao, C., Tang, F., Wang, W. & Zhu, X. Dynamin-like protein 1 cleavage by calpain in Alzheimer's disease. *Aging Cell* **18**, e12912 (2019).

66. Cereghetti, G. M. et al. Dephosphorylation by calcineurin regulates translocation of Drp1 to mitochondria. *Proc. Natl Acad. Sci. USA* **105**, 15803–15808 (2008).

67. Wang, L. et al. Mitofusin 2 regulates axonal transport of calpastatin to prevent neuromuscular synaptic elimination in skeletal muscles. *Cell Metab.* **28**, 400.e8–414.e8 (2018).

68. Wisniewski, J. R., Zougman, A., Nagaraj, N. & Mann, M. Universal sample preparation method for proteome analysis. *Nat. Methods* **6**, 359–362 (2009).

69. Keller, A., Nesvizhskii, A. I., Kolker, E. & Aebersold, R. Empirical statistical model to estimate the accuracy of peptide identifications made by MS/MS and database search. *Anal. Chem.* **74**, 5383–5392 (2002).

70. Nesvizhskii, A. I., Keller, A., Kolker, E. & Aebersold, R. A statistical model for identifying proteins by tandem mass spectrometry. *Anal. Chem.* **75**, 4646–4658 (2003).

## Acknowledgements

This study is supported by grants from the US National Institutes of Health (R01AG065240, R01NS115903, R21NS107897 to X.Q.), Dr. Ralph and Marian Falk Medical Research Trust-Catalyst and -Transformative Awards, a Harrington Discovery Institute Rare Disease Scholar Award to X.Q. and D.J.A., the Mount Sinai Health Care Foundation, Thomas F. Peterson, Jr., Case Western Reserve University School of Medicine, and the Case Comprehensive Cancer Center (P30 CA043703). The mouse genotype study was supported by a P30 core grant for vision research (NIH 5P30EY011373). We thank Dr. Eliezer Masliah of the University of California at San Diego for providing frozen postmortem brain samples of patients with HD and normal subjects. We are grateful to the Case Western Reserve University Small-Molecule Drug Development and Proteomics Cores for technical support. We thank Dr. Florence Wagner (Broad Institute) for providing reagents BRD0320, BRD3731, and BRD0705.

## Author contributions

D.H. performed most of the cell culture, animal model, and patient sample experiments and also drafted the manuscript. X.S. helped maintain mouse lines and collect tissue samples. M.T., A.M., and Y.F. performed HTS and chemical genetic screening studies. B.W. and K.L. performed proteomics analysis and analyzed data; D.J.A. designed and supervised HTS and GSK3 inhibitor evaluation studies and also edited the manuscript. X.Q. conceived, designed, supervised all studies, and edited the manuscript.

## Competing interests

The authors declare no competing interests.

**18**     NATURE COMMUNICATIONS | (2021)12:5305 | https://doi.org/10.1038/s41467-021-25651-y | www.nature.com/naturecommunications
