## [Peer Review File · Nature Communications]

Reviewers' Comments:

Reviewer #1:

Remarks to the Author:

Hu et al manuscript described a small molecule compound CHIR99021 identified from an HTS screening showed a protective effect on mutant HTT toxicity in multiple models of Huntington's disease (HD), including human iPSCs, mouse striatal-like cell model and two mouse models. Authors explored the protective mechanism of this compound in HD cell models and discovered that the protective effect of CHIR compound is mediated by stabilizing endogenous calpain inhibiting protein Calpastatin (CAST) which is depleted in HD human caudate putamen and HD models rather than its traditional role of this compound as a GSK3 inhibitor. Furthermore, they demonstrated that stabilizing CAST levels by this compound also prevented mitochondrial fission protein Drp1 activation thereby preserved the mitochondrial function in HD models. The findings reported in this manuscript is significant, as there is no disease modifying therapy for HD, a small molecule compound with a novel mechanism could be a promising candidate for therapeutic development for this devastating disease. Although it was reported previously that overexpression of CAST protected HD mice, but the findings which a brain-penetrable small molecule compound stabilized CAST levels and exhibited distinguished mechanism of protection in HD remain innovative. In addition, the mechanistic study suggested that this compound specifically reduced ubiquitination of CAST in HD condition while no effect on CAST levels in normal cells, and preservation of CAST levels by the compound in HD condition is mediated by attenuating its degradation via proteasome pathway.

There are some minor comments:

1. Q7 cell results should be included in Figure 1E to show readers the amplitude of these parameters compromised in Q111 cells, and Q7 group values should be used for normalization as authors did in other panels.
2. Page 5, description of EC50 values, it looks like a typo, 0.3 to < (instead of >) 5uM?
3. All the cell toxicity related measures, such as LDH release and ATP levels, are influenced by cell density, Q7 and Q111 cells have different proliferation rates, authors may add description in methods on how the cell density was controlled in these assays.
4. In animal studies, author reported that the compound was given to R6/2 mice 5 days a week and to YAC128 mice every other day, the rationale of these drug administration regimen should be described.
5. When describing the open field results, authors used "motor function" instead of "locomotor activity" which is more precisely describe the open field measures. Have rotarod tests been done in R6/2 mice as did in YAC128, rotarod is better to measure motor coordination. How about the body weight in treated animals? did the compound affect body weight in both HD models? Body weight also contribute to the performance of rotarod tests.
6. Author may add discussion on possible mechanism of how this compound selectively prevents ubiquitination of CAST in HD condition?

Reviewer #2:

Remarks to the Author:

In this manuscript, Hu et al. describe that the stabilization of calpastatin (CAST) by a small molecule, CHIR99021 (CHIR), is protective in Huntingtin Disease-associated models. Specifically, CHIR enhances mitochondrial function, improves mitochondrial phenotypes in cellular models of HD (including striatal neurons), and ameliorates defects in R6/2 and YAC128 HD mice models. CHIR99021 suppresses the degradation of calpastatin and consequently the markers of calpain activation. In addition, CHIR99021 suppresses mitochondrial fragmentation driven by Dpr1. Though the proposed mechanism is interesting and the authors identify the CAST pathway as a target of CHIR in an unbiased manner, additional experiments and controls are needed to further strengthened the main conclusions.

- 1) "Notably, CHIR99021 treatment in HdhQ111 cells improved MMP to levels observed in wild-type HdhQ7 cells"

S1C: Please provide the MMP measurements in wild-type cells side-by-side to substantiate this conclusion.

2) Does CHIR affect mitochondrial respiration in wild-type cells? i.e. without a pathogenic background, does treatment with CHIR alter the baseline mitochondrial homeostasis?

3) Fig 1G-J: Please provide a dose-response of the protection of human neurons with CHIR treatment. It is not clear how the concentration of CHIR is chosen in each experiment 3uM in cell lines vs 1 uM for 5 days in human neurons.

4) The authors cite a compound report for data on the brain penetration of CHIR (reference 29). The details of the pharmacodynamic data are not immediately clear in this report. To substantiate the main claim that CHIR protects in HD models and to validate the dosing regimen used, the authors need to show (or reference) data on the extent of BBB penetration of CHIR, the concentration of the drug in the free fraction in the brain, and how this concentration compares to the potency of CHIR as measured in the cellular assays.

What is the reason for different dosing regimens in Fig 2A vs 2G between the two HD models?

5) What do the individual values represent in IHC quantification images? For example, in Fig 2E, 6 mice/group were used, however, the graphs have ~20 data points/group. It will be better if the final analysis is performed on a per animal basis.

6) Does CHIR treatment reduce Htt aggregates as measured by biochemistry?

7) Does the activity of calpain as measured by the enzymatic assay show a difference between the wild-type vs HD mice, and by CHIR treatment? In other words, does the increased protein level of calpain translate into increased calpain activity? Cleaved spectrin is a surrogate marker of calpain activity but may not be specific since spectrin can also be cleaved by other proteases.

8) What is the knockdown efficiency of CAST in Figure 5A? There is still some protection remaining with CAST knockdown. CAST KO cells could give cleaner results.

The authors use stressed HdhQ7 cells to show the requirement of CAST for the beneficial effects of CHIR. On the other hand, Figure 1 uses HdhQ111 or HD neuronal cells to show the rescue of MMP, cellular viability, and neuronal axons by CHIR. Is CAST required in HD cells for the beneficial effects of CHIR? In other words, are the beneficial effects of CHIR lost/reduced in HdhQ111 cells and in HD neurons upon knockout of CAST? Same comment for Fig. 6E. Similarly, are the effects of CHIR dependent on calpain?

9) Is the reduction in the protein level of CAST upon CHX treatment (Fig S5C) prevented by proteasome inhibition? Please use additional proteasome inhibitors to substantiate the data on the requirement for proteasome for CAST degradation (e.g. lactacystin. MG132 is a non-specific proteasome inhibitor). Please provide the data on calpain levels and activity in Fig 5F.

10) The authors could speculate on the MOA of CHIR further in the discussion since it seems the molecule does not induce accumulation of ubiquitinated CAST but rather acts upstream of CAST ubiquitination. What are the ligases/DUBs known to regulate CAST? Could these be the direct targets of CHIR?

11) In certain figures, the authors compare the experimental group to a normalized (control) group where there is no variability. It may be better to use statistical tests that report if the mean of the experimental groups is different than 1 in these cases.

Reviewer #3:

Remarks to the Author:

In the manuscript by Xin Qi and colleagues, the authors have characterised a small molecule,

originally designed as a GSK3 inhibitor, CHIR99021, to have neuroprotective properties by enhancing mitochondrial function. The authors observed beneficial effects on mitochondrial phenotypes, and this drug enhances cell viability in multiple models of Huntington's disease (HD). The authors could not attribute this effect to the compound's classical mechanism of action via GSK3 inhibition, but rather through an off-target effect. The authors provide evidence for CHIR99021 to have a partial effect on proteasomal proteolysis, leading to stabilisation of calpastatin (CAST) as this interferes with Calpain-1 protease activation and subsequent protein degradation in neuronal cells.

The study includes aspects of discovery (HTS) and a thorough biological characterisation of small molecule intervention in vitro and in vivo. Whereas these parts of the study appear to be done well, there are some concerns about the proposed mechanism of action (off-target) effect that should be addressed as described below.

Major points:

The proposed alternative mechanism of the small molecule compound CHIR99021, namely to suppress degradation of CAST and Drp1 via selective inhibition of proteasomal degradation, is potentially interesting, but is riddled with concerns. Inhibition of the proteasome, although possible to achieve in a selective manner through targeting individual catalytic subunits of either immun- or normal proteasome types, usually affects ubiquitin mediated degradation of proteins in a global manner, perhaps with a greater effect on the turn-over of short-lived versus long-lived proteins. So, how do the authors explain selective inhibition of these proteins versus the "bulk" of proteins degraded by the proteasome? As Figure 5E shows, there is moderate CAST ubiquitination upon CHIR99021 treatment as compared to MG132 proteasome inhibitor, and the stabilisation of CAST appears marginal (Figure 5F). This raises some serious concerns about the author's conclusions about an alternative mechanism of action of CHIR99021.

Minor points:

Label-free proteomics method. The sample amounts mentioned in the method section should be checked carefully. For example, on page 26, 'peptide digests (320 mg) were loaded onto a column in an 8 μ L injection volume' is simply not possible.

On page 27, for the power analysis carried out, no details have been provided, and the approach used should be referenced.

Whereas the abstract and parts of the introduction & results are easy to read, other parts of the manuscript, in particular the methods section, deserve improvements in editing and English language.

Response letter

RE: Small-molecule suppression of calpastatin degradation reduces neuropathology in models of Huntington's disease (NCOMMS-20-48977)

We thank the reviewers for their constructive comments and criticisms. In the following sections, we provide point-by-point responses to these comments. Because of additional experiments and changes requested, we made a number of changes to the revised manuscript. These are described in detail below. Figures, results, figure legends, discussion, and methods are added at appropriate places of the revised manuscript; All newly added sentences are underlined and marked in blue in the manuscript. We believe these changes have significantly improved our manuscript and we hope the editor and reviewers will now find the revised manuscript suitable for publication.

Response to Reviewers' Comments:

Reviewer #1:

Hu et al manuscript described a small molecule compound CHIR99021 identified from an HTS screening showed a protective effect on mutant HTT toxicity in multiple models of Huntington's disease (HD), including human iPSCs, mouse striatal-like cell model and two mouse models. Authors explored the protective mechanism of this compound in HD cell models and discovered that the protective effect of CHIR compound is mediated by stabilizing endogenous calpain inhibiting protein Calpastatin (CAST) which is depleted in HD human caudate putamen and HD models rather than its traditional role of this compound as a GSK3 inhibitor. Furthermore, they demonstrated that stabilizing CAST levels by this compound also prevented mitochondrial fission protein Drp1 activation thereby preserved the mitochondrial function in HD models.

The findings reported in this manuscript is significant, as there is no disease modifying therapy for HD, a small molecule compound with a novel mechanism could be a promising candidate for therapeutic development for this devastating disease. Although it was reported previously that overexpression of CAST protected HD mice, but the findings which a brain-penetrable small molecule compound stabilized CAST levels and exhibited distinguished mechanism of protection in HD remain innovative. In addition, the mechanistic study suggested that this compound specifically reduced ubiquitination of CAST in HD condition while no effect on CAST levels in normal cells, and preservation of CAST levels by the compound in HD condition is mediated by attenuating its degradation via proteasome pathway.

There are some minor comments:

1. Q7 cell results should be included in Figure 1E to show readers the amplitude of these parameters compromised in Q111 cells, and Q7 group values should be used for normalization as authors did in other panels.

Response: In response to these comments, we included the quantification of basal respiration, maximal respiration, and ATP production in HdhQ7 cells (Figure 1E). All groups are now normalized to HdhQ7 cells as shown in histograms in the revised manuscript.

2. Page 5, description of EC50 values, it looks like a typo, 0.3 to < (instead of >) 5uM?

Response: We have corrected this error.

3. All the cell toxicity related measures, such as LDH release and ATP levels, are influenced by cell

density, Q7 and Q111 cells have different proliferation rates, authors may add description in methods on how the cell density was controlled in these assays.

Response: To measure mitochondrial respiration in HdhQ7 and HdhQ111 cells, all results were normalized to total protein content in each group as described in the Methods. To measure LDH release, we added the following description in Methods section:

“Relative LDH release was calculated as: $LDH_{medium}/(LDH_{medium} + LDH_{cell})$ and then normalized to HdhQ7 cells data.” (page 23 of revised manuscript)

4. In animal studies, author reported that the compound was given to R6/2 mice 5 days a week and to YAC128 mice every other day, the rationale of these drug administration regimen should be described.

Response: Using HD R6/2 mice which exhibit rapid and severe disease symptoms, we aimed to assess whether administration of CHIR99021 at an early time-point would prevent the progression of HD neuropathology and motor dysfunction in a relatively short period. Thus, mice were given CHIR99021 at a regimen of 5 days/week. YAC128 mice progressively develop HD-like symptoms starting from six months old, neurodegeneration at nine months, and hypokinesia at twelve months. These mice more closely recapitulate the chronic neuropathology in human HD. Similar to the chronic treatment regimen of tetrabenazine (the only Food and Drug Administration approved drug to ameliorate HD-related symptoms for patients) in YAC128 mice (Wang *et al.* Molecular Neurodegeneration, 2010), the intraperitoneal injection of CHIR99021 was given 3 days per week, starting at 9 months to mimic post-symptomatic treatment and long-term intervention in the clinic. By performing such treatment regimens, we assessed if the chronic administration of CHIR99021 could reduce HD-related symptoms in YAC128 mice. In response to the reviewer’s comments, we added the following sentences to the results section to extrapolate our CHIR99021 administration rationale in our mouse models.

“Because CHIR99021 passes the blood-brain barrier²², we intraperitoneally (i.p.) treated HD R6/2 and WT mice with the molecule (10 mg/kg/day, five days/week) starting at 6 weeks old (Fig. 2A) to determine whether CHIR99021 administration prevented rapid and severe progression of HD-related pathology.” (page 7 in the revised manuscript)

“Similar to a treatment regimen in previous work²⁷, we treated YAC128 mice and WT littermates with CHIR99021 and vehicle starting at 9 months old, 3 months after detectable motor deficits appear in YAC128 mice (Fig. 2G). We ended treatment at 12 months old. Initiating treatment at 9 months better mimicked post-symptomatic, chronic treatments administered to HD patients²⁸.” (page 7 in the revised manuscript)

5. When describing the open field results, authors used “motor function” instead of “locomotor activity” which is more precisely describe the open field measures. Have rotarod tests been done in R6/2 mice as did in YAC128, rotarod is better to measure motor coordination. How about the body weight in treated animals? did the compound affect body weight in both HD models? Body weight also contribute to the performance of rotarod tests.

Response: R6/2 mice are fragile and exhibit high lethality due to handling stress from 10 weeks onwards (Mangiarini *et al.* 1996). The severe disease development in these mice did not permit rotarod tests which require three consecutive days of training and multiple handling. Alternatively, we applied clasping behavior analyses to assess motor deficits in these animals (Figure 2D in the original manuscript). In addition, we corrected “motor function” as “locomotor activity” in the manuscript.

In response to reviewer's comments, we provided body weight data for both R6/2 and YAC128 mice (Fig. S2H). We observed no significant differences in these parameters between vehicle and CHIR99021-treated HD mice. The below sentence is added to the Results section (page 7-8 in the revised manuscript)

“Importantly, this dosing regimen was not toxic: CHIR99021 treatments had no effects on behavioral status, body weight, and survival rates of WT mice over the 6-week treatment period (Fig. 2, Fig. S2).”

“Notably, CHIR99021 treatment exerted no effects on YAC128 mice body weight, over a wide age range (Fig. S2H).”

6. Author may add discussion on possible mechanism of how this compound selectively prevents ubiquitination of CAST in HD condition?

Response: In response to reviewer's comments, we have added the following discussion in the revised manuscript (page 17)

“To our knowledge, ours is the first study to reveal the UPS-mediated degradation of CAST. In HdhQ111 cells, the CHX-induced degradation of CAST was prevented by treating cells with selective proteasome inhibitors, suggesting a high turn-over rate for the CAST protein mediated by the proteasome. Although CHIR99021 treatment prevented CAST loss in HdhQ111 cells, it did not improve MCL1 protein levels - a protein degraded by the UPS pathway⁴⁴, consistent with the observation that CHIR99021 did not affect proteasome activity. These lines of evidence suggest that CHIR99021 may selectively modulate CAST degradation rather than “bulk” protein degradation by the proteasome. CAST can be cleaved and degraded by certain proteases, including calpain⁵⁶. However, our results excluded the possibility that CHIR99021 directly affected calpain enzymatic activity. Because the accumulation of CAST ubiquitination was abolished upon CHIR99021 treatment in HdhQ111 cells, we speculate that CHIR99021 might inhibit ubiquitin ligases or stimulate deubiquitinases that regulate CAST ubiquitination and degradation. Therefore, CHIR99021 may serve as “a CAST-stabilizer” to suppress calpain-mediated cellular injury in HD. Future investigations into the regulation of CAST ubiquitination in HD are warranted to elucidate detailed pathways where CHIR99021 stabilizes CAST.”

Reviewer #2:

In this manuscript, Hu et al. describe that the stabilization of calpastatin (CAST) by a small molecule, CHIR99021 (CHIR), is protective in Huntingtin Disease-associated models. Specifically, CHIR enhances mitochondrial function, improves mitochondrial phenotypes in cellular models of HD (including striatal neurons), and ameliorates defects in R6/2 and YAC128 HD mice models. CHIR99021 suppresses the degradation of calpastatin and consequently the markers of calpain activation. In addition, CHIR99021 suppresses mitochondrial fragmentation driven by Drp1. Though the proposed mechanism is interesting and the authors identify the CAST pathway as a target of CHIR in an unbiased manner, additional experiments and controls are needed to further strengthened the main conclusions.

1) “Notably, CHIR99021 treatment in HdhQ111 cells improved MMP to levels observed in wild-type HdhQ7 cells”

S1C: Please provide the MMP measurements in wild-type cells side-by-side to substantiate this conclusion.

Response: We provided mitochondrial membrane potential (MMP) measurements from HdhQ7 cells (Figure 1A) in the revised manuscript. CHIR99021 treatment had no effect on MMP in WT-HdhQ7 cells at doses used for HdhQ111 mutant cells.

2) Does CHIR affect mitochondrial respiration in wild-type cells? i.e. without a pathogenic background, does treatment with CHIR alter the baseline mitochondrial homeostasis?

Response: To address the reviewer's comment, we provided mitochondrial respiration measurements in HdhQ7 cells treated with CHIR99021 (Figure S1D). CHIR99021 treatment had no effect on mitochondrial respiration in wildtype HdhQ7 cells. The following sentence was added to the revised manuscript (page 6). In addition, the figure legend was also added.

“In contrast, CHIR99021 had no effects on mitochondrial respiration in HdhQ7 cells (Fig. S1D).”

3) Fig 1G-J: Please provide a dose-response of the protection of human neurons with CHIR treatment. It is not clear how the concentration of CHIR is chosen in each experiment 3uM in cell lines vs 1 uM for 5 days in human neurons.

Response: We provided data for the dose-dependent protection effects of CHIR99021 (0.5–3 μM) on MMP and cell viability in HD neurons differentiated from patient iPS cells (Figure S1E in the revised manuscript). A corresponding figure legend was also added.

In HdhQ111 striatal cells, the EC_{50} of CHIR99021 was $\sim 3 \mu\text{M}$. Thus, we used this dosage (3 μM for two days) for studies. We previously developed peptide inhibitors, P110, HV-3, and DA1, which blocked the protein-protein interaction of Drp1/Fis1, VCP/Htt, and Drp1/ATAD3A, respectively, improved mitochondrial function, and exhibited neuroprotection in various HD models (*Guo et al., J Clin Invest, 2013; Guo et al., Nat Commun 2016; Zhao et al., Nat Commun 2019*). Treatment with these peptide inhibitors (1 $\mu\text{M}/\text{day}$ for 5 days) after 20 days neuronal differentiation improved the survival of HD neurons from patient iPS cells. Thus, we referred to this treatment regimen in our current study.

The following sentences were added to Results section (page 6). The figure legend for Figure S1E was also added.

“Dose-dependent CHIR99021 protection toward MMP and cell viability was also observed in neurons differentiated from HD patient iPS cells (Fig. S1E).”

4) 1. The authors cite a compound report for data on the brain penetration of CHIR (reference 29). The details of the pharmacodynamic data are not immediately clear in this report. To substantiate the main claim that CHIR protects in HD models and to validate the dosing regimen used, the authors need to show (or reference) data on the extent of BBB penetration of CHIR, the concentration of the drug in the free fraction in the brain, and how this concentration compares to the potency of CHIR as measured in the cellular assays. What is the reason for different dosing regimens in Fig 2A vs 2G between the two HD models?

Response: We thank the reviewer for these constructive comments. We included a study (Pan *et al.*, 2011) where CHIR99021 pharmacokinetics were well characterized after intraperitoneal (i.p.) injection (see new reference 22 in the revised manuscript). In this study, one hour after injection at 12.5 mg/kg, the maximum brain CHIR99021 concentration (C_{max}) was 0.16 μM and brain area under the curve (AUC) data was 0.15 $\mu\text{M}/\text{kg}\cdot\text{hr}$. These pharmacokinetic data clearly demonstrated brain penetration of CHIR99021 following i.p. injection.

We acknowledge that brain CHIR99021 concentrations are somewhat lower than effective concentrations in our cell-based assays. These may be due to several reasons. The cellular EC₅₀ may be overestimated which may result from the use of cell lines and the sensitivity of assays used. Our treatment regimen for CHIR99021 in HD mice (once/day over 6 weeks in HD R6/2 mice and once every other day over 3 months in YAC128 mice) may have led to accumulated brain concentrations of the compound, and higher than a single dose as used by Pan *et al.* In addition, blood-brain-barrier breakdown has been reported as an early pathological event in both HD R6/2 and YAC128 mice (Di Pardo *et al.*, *Sci Rep* 2017; Franciosi *et al.*, *Neurobiol Dis* 2011) which may also lead to a higher concentration of CHIR99021 in HD mice brains following administration. Future studies on the measurement of brain CHIR99021 concentrations in HD mice will improve the translational application of small molecules and derived analogs. Finally, CHIR99021 has been shown to induce positive effects in other neurological disease models, including bipolar disorder (Pan *et al.*, *Neuropsychopharmacology* 2011) further corroborating the neuroprotective effects following CHIR99021 treatment.

In terms of different treatment regimens in HD R6/2 and YAC128 mice, please refer to the response to reviewer 1, comment #4.

5) What do the individual values represent in IHC quantification images? For example, in Fig 2E, 6 mice/group were used, however, the graphs have ~20 data points/group. It will be better if the final analysis is performed on a per animal basis.

Response: Per the reviewer's suggestion, we re-analyzed our data and reconstructed histograms on a per animal basis (Figure 2E, 2F, 2I, and 2J) in the revised manuscript. The number of animals in each group was also added to figure legends. (For Fig 2E, WT: $n = 3$, R6/2 veh: $n = 3$, R/62 CHIR99021: $n = 5$; for Fig 2F, R/62 veh: $n = 3$, R/62 CHIR99021: $n = 4$; for Fig 2I, $n = 7$ for each group; and for Fig 2J, $n = 9$ for each group).

6) Does CHIR treatment reduce Htt aggregates as measured by biochemistry?

Response: In response to reviewer's comment, we performed dot blot and western blot to examine Htt aggregates using an anti-Htt antibody (clone EM48 recognizes Htt aggregates in mouse tissue) in the triton-insoluble fraction from mouse striatum tissue. Representative blots and associated quantification data are now provided (Figure S2B, C, F). These findings were consistent with immunohistochemistry showing that CHIR99021 treatment reduced Htt aggregates in the HD mouse brain. Figure legends were updated in the revised supplemental information.

7) Does the activity of calpain as measured by the enzymatic assay show a difference between the wild-type vs HD mice, and by CHIR treatment? In other words, does the increased protein level of calpain translate into increased calpain activity? Cleaved spectrin is a surrogate marker of calpain activity but may not be specific since spectrin can also be cleaved by other proteases.

Response: In response to reviewer's comments, we determined calpain enzymatic activity in WT and HD mice with or without CHIR99021 treatment. We showed that calpain activity was substantially increased in both HD R6/2 and YAC128 mouse brains, and CHIR99021 treatment suppressed calpain activity. Given that CHIR99021 does not directly suppress calpain enzymatic activity *in vitro*, we speculated the CHIR99021 suppression of calpain enzymatic activity in HD mice was indirect, most likely resulting from elevated CAST protein levels. These results are now provided (Figure S4F and G) in the revised manuscript. We agree with the reviewer's comment that α -spectrin can be cleaved by other proteases. In the original manuscript, we also examined the cleavage of the p35 protein, a specific substrate of calpain (Kusakawa *et al.*, *J Biol Chem* 2000) in both R62 and YAC128 mice upon CHIR99021 treatment

(Figure S4E). The following sentences were added to the revised manuscript (page 11). Figure legends were also added.

“Additionally, we directly measured calpain activities in the brains of R6/2 and YAC128 mice. As predicted by upregulated calpain I protein levels observed earlier, calpain enzymatic activity was substantially higher in both R6/2 and YAC128 mice (Fig. S4F, G). CHIR99021 treatment diminished calpain activity to near WT levels (Fig. S4F, G), consistent with its ability to abrogate calpain substrate cleavage.”

8) What is the knockdown efficiency of CAST in Figure 5A? There is still some protection remaining with CAST knockdown. CAST KO cells could give cleaner results. The authors use stressed HdhQ7 cells to show the requirement of CAST for the beneficial effects of CHIR. On the other hand, Figure 1 uses HdhQ111 or HD neuronal cells to show the rescue of MMP, cellular viability, and neuronal axons by CHIR. Is CAST required in HD cells for the beneficial effects of CHIR? In other words, are the beneficial effects of CHIR lost/reduced in HdhQ111 cells and in HD neurons upon knockout of CAST? Same comment for Fig. 6E. Similarly, are the effects of CHIR dependent on calpain?

Response: We thank the reviewer for this constructive comment/suggestion. We knocked down CAST by lentiviral-shRNA in HdhQ111 striatal cells and HD patient neurons and determined the CAST-dependent protection of CHIR99021. The knock-down efficiency of CAST (> 90%) in HdhQ111 cells and HD neurons was examined by western blotting (Figs. 5A and 5F) in the revised manuscript, respectively. Furthermore, CAST knock-down abolished the protection of CHIR99021 on MMP and cell viability in HdhQ111 cells (Fig. 5B, C in the revised manuscript). CHIR99021 treatment did not rescue mitochondrial morphology in CAST-deficient HdhQ111 cells (Fig. 7E, F in the revised manuscript). Similarly, the protection of CHIR99021 in HD neurons from patient iPSC cells was abolished upon CAST knock-down (Fig. 5G–J in the revised manuscript).

In parallel, we knocked down calpain using lentiviral-shRNA in HdhQ111 cells. The calpain knock-down efficiency (> 90%) is shown (Fig. 5A in the revised manuscript). Calpain knock-down significantly improved MMP and cell viability in HdhQ111 cells, whereas CHIR99021 administration had no additional protective effects on these parameters in calpain-deficient HdhQ111 cells (Fig. 5D and E). CHIR99021 also had no additional effects on the reduction of mitochondrial fragmentation in calpain-deficient HdhQ111 striatal cells (Fig. 7G and H in the revised manuscript). These findings collectively suggested that the CAST/calpain signaling pathway mediated CHIR99021 mitochondrial protection in HD cells.

The following sentences were added to the Results section in the revised manuscript (page 12). The figure legends were also added.

“To determine whether the protective effects of CHIR99021 in our HD models depended on the presence of CAST, we knocked down CAST using short hairpin RNA (shRNA) in HdhQ111 cells (Fig. 5A) and determined MMP and cell viability (Fig. 5B, C). In the presence of the control shRNA (shCon), CHIR99021 greatly improved MMP and cell viability in these cells. In contrast, CAST knock-down abolished the protective effects of CHIR99021 toward MMP and cell viability (Fig. 5B, C). In parallel, we also knocked down calpain I by shRNA (shCAPN) (Fig. 5A). This knock-down significantly improved MMP and reduced cell death in HdhQ111 cells (Fig. 5D, E). Treatment with CHIR99021 had no additional effects on MMP and cell viability in HdhQ111 cells expressing shCAPN; the effects were similar to calpain knock-down or CHIR99021 treatment-alone effects (Fig. 5D, E).” (page 12)

“Next, to examine whether CAST mediated the neuronal protection of CHIR99021 in human HD, we downregulated CAST using lentiviral shRNA in neurons derived from HD patient iPSCs (Fig. 5F).

Consistent with our earlier findings (Fig. 1F, G), CHIR99021 treatment significantly improved MMP and cell viability in HD patient iPS cells-derived neurons infected with control shRNA (Fig. 5G). However, in patient iPS cells-derived neurons infected with CAST shRNA (shCAST), CHIR99021 had no protective effects on MMP or cell viability (Fig. 5G). Moreover, CHIR99021 treatment failed to rescue the dendritic and axonal shortening in shCAST-expressing neurons derived from HD patient iPS cells (Fig. 5H–J). Collectively, our results indicate that CHIR99021-mediated mitochondrial protection in HD cells requires the presence of CAST.” (page 12)

“Furthermore, CHIR99021 treatment failed to reduce mitochondrial fragmentation in HdhQ111 striatal cells in the presence of CAST shRNA (Fig. 7E, F) or calpain shRNA (Fig. 7G, H).” (page 14)

9) Is the reduction in the protein level of CAST upon CHX treatment (Fig S5C) prevented by proteasome inhibition? Please use additional proteasome inhibitors to substantiate the data on the requirement for proteasome for CAST degradation (e.g. lactacystin. MG132 is a non-specific proteasome inhibitor). Please provide the data on calpain levels and activity in Fig 5F.

Response: In response to reviewer’s comments, we generated western blots for CAST upon CHX treatment followed by proteasome inhibition by MG132, lactacystin or epoxomicin in HdhQ111 cells (Figure S6C–E in the revised manuscript). Proteasome inhibition suppressed the reduction in CAST protein levels upon CHX treatment, suggesting a proteasome-mediated degradation of CAST in HdhQ111 cells.

We also generated western blots of CAST and calpain in HdhQ111 cells treated with lactacystin or epoxomicin (Fig. 6A, B) or MG132 (Fig. S6F) in the presence of CHIR99021. Treatment with these proteasome inhibitors improved CAST protein levels. Co-treatment with CHIR99021 had no additional effects on protein levels under the same conditions. In parallel, we examined calpain activity in HdhQ111 cells treated with lactacystin, epoxomicin or MG132 in the presence of CHIR99021 (Fig. 6C; Fig. S6G). Treatment with these proteasome inhibitors together with CHIR99021 suppressed calpain activity, which was consistent with increased CAST protein levels.

The following sentences are added to the Results section (page 13) in the revised manuscript.

“The loss of CAST in CHX-treated HdhQ111 cells was prevented by the addition of multiple structurally-unrelated proteasome inhibitors, including lactacystin (Lac), epoxomicin (epo), or MG132 (Fig. S6C–E). Interestingly, while these additions restored CAST protein levels in HdhQ111 cells, co-treatment with CHIR99021 did not generate additional improvements under the same conditions (Fig. 6A, B; Fig. S6F). Also, CHIR99021 treatment reduced calpain I protein levels and suppressed its activity in HdhQ111 cells in the presence of these inhibitors (Fig. 6A–C; Fig. S6F, G). In addition, treatment with MG132 induced a substantial accumulation of ubiquitinated CAST in HdhQ111 striatal cells, which was abolished by CHIR99021 treatment (Fig. 6D). These results collectively suggested that CAST had a shorter half-life in HD cells than WT cells; CAST was degraded via the UPS, with CAST ubiquitination and degradation both prevented by CHIR99021. CHIR99021 treatment had no effect on proteasome activity *in vitro* and in HdhQ7/111 cells (Fig. S6H, I) nor on MCL1 protein levels, a well-known proteasome substrate⁴⁴ (Fig. 6A, B; Fig. S6F), thereby excluding a global inhibition of CHIR99021 on proteasome activity. Taken together, our results led us to propose a model that CHIR99021 treatment prevents the UPS-mediated degradation of CAST in the context of HD.”

10) The authors could speculate on the MOA of CHIR further in the discussion since it seems the molecule does not induce accumulation of ubiquitinated CAST but rather acts upstream of CAST ubiquitination. What are the ligases/DUBs known to regulate CAST? Could these be the direct targets of CHIR?

Response: We thank reviewer for the constructive suggestion. In response to reviewer's comments, we have added the following discussion in the revised manuscript.

“To our knowledge, ours is the first study to reveal the UPS-mediated degradation of CAST. In HdhQ111 cells, the CHX-induced degradation of CAST was prevented by treating cells with selective proteasome inhibitors, suggesting a high turn-over rate for the CAST protein mediated by the proteasome. Although CHIR99021 treatment prevented CAST loss in HdhQ111 cells, it did not improve MCL1 protein levels - a protein degraded by the UPS pathway⁴⁴, consistent with the observation that CHIR99021 did not affect proteasome activity. These lines of evidence suggest that CHIR99021 may selectively modulate CAST degradation rather than “bulk” protein degradation by the proteasome. CAST can be cleaved and degraded by certain proteases, including calpain⁵⁶. However, our results excluded the possibility that CHIR99021 directly affected calpain enzymatic activity. Because the accumulation of CAST ubiquitination was abolished upon CHIR99021 treatment in HdhQ111 cells, we speculate that CHIR99021 might inhibit ubiquitin ligases or stimulate deubiquitinases that regulate CAST ubiquitination and degradation. Therefore, CHIR99021 may serve as “a CAST-stabilizer” to suppress calpain-mediated cellular injury in HD. Future investigations into the regulation of CAST ubiquitination in HD are warranted to elucidate detailed pathways where CHIR99021 stabilizes CAST.” (page 17)

11) In certain figures, the authors compare the experimental group to a normalized (control) group where there is no variability. It may be better to use statistical tests that report if the mean of the experimental groups is different than 1 in these cases.

Response: Per the reviewer's suggestions, we compared experimental group means with control group means and provided *p*-value in figures.

Reviewer #3 (Remarks to the Author):

In the manuscript by Xin Qi and colleagues, the authors have characterised a small molecule, originally designed as a GSK3 inhibitor, CHIR99021, to have neuroprotective properties by enhancing mitochondrial function. The authors observed beneficial effects on mitochondrial phenotypes, and this drug enhances cell viability in multiple models of Huntington's disease (HD). The authors could not attribute this effect to the compound's classical mechanism of action via GSK3 inhibition, but rather through an off-target effect. The authors provide evidence for CHIR99021 to have a partial effect on proteasomal proteolysis, leading to stabilisation of calpastatin (CAST) as this interferes with Calpain-1 protease activation and subsequent protein degradation in neuronal cells.

The study includes aspects of discovery (HTS) and a thorough biological characterisation of small molecule intervention *in vitro* and *in vivo*. Whereas these parts of the study appear to be done well, there are some concerns about the proposed mechanism of action (off-target) effect that should be addressed as described below.

Major points:

The proposed alternative mechanism of the small molecule compound CHIR99021, namely to suppress degradation of CAST and Drp1 via selective inhibition of proteasomal degradation, is potentially interesting, but is riddled with concerns. Inhibition of the proteasome, although possible to achieve in a selective manner through targeting individual catalytic subunits of either immuno-

or normal proteasome types, usually affects ubiquitin mediated degradation of proteins in a global manner, perhaps with a greater effect on the turn-over of short-lived versus long-lived proteins. So, how do the authors explain selective inhibition of these proteins versus the “bulk” of proteins degraded by the proteasome? As Figure 5E shows, there is moderate CAST ubiquitination upon CHIR99021 treatment as compared to MG132 proteasome inhibitor, and the stabilisation of CAST appears marginal (Figure 5F). This raises some serious concerns about the author’s conclusions about an alternative mechanism of action of CHIR99021.

Response: We thank the reviewer for these constructive comments. Firstly, our model does not suggest CHIR99021 directly modulates the proteasome. We added new data examining the direct effects of CHIR99021 on proteasome activity in HdhQ7 and HdhQ111 cells by enzymatic assay. The data showed that CHIR99021 treatment did not affect proteasome activity in HdhQ7 and HdhQ111 cells (Figure S6H and I in the revised manuscript). Secondly, we determined the effects of CHIR99021 on MCL1 protein levels - MCL1 is degraded by the ubiquitin-proteasome pathway. While treatment with the proteasome inhibitors, lactacystin or epoxomicin or MG132, significantly elevated MCL1 levels in HdhQ111 cells, CHIR99021 had no effect on these levels (Fig. 6A, B; Fig. S6F). These findings consistently suggest CHIR99021 does not affect global proteasome degradation.

Instead, our model suggests CHIR99021 more than likely affects CAST ubiquitination, which dictates its degradation by the proteasome. In support of this, Fig. 5E (now Fig. 6D due to new data) indicates that accumulation of ubiquitinated CAST due to proteasome inhibition was massively reduced by CHIR99021. Also, CAST protein levels were clearly and reproducibly upregulated following CHIR99021 treatment (Fig 6A and 6B; cf. lane 2 and 4 in each panel). Moreover, CHIR99021 exerted no additional effects on CAST protein levels in the presence of the three proteasome inhibitors (lactacystin, epoxomicin, and MG132). Additionally, an independent quantitative proteomics analysis corroborated a 2-fold upregulation of CAST protein levels following CHIR99021 treatment.

Notably, ubiquitination and deubiquitination are highly regulated processes specific for individual proteins, including CAST. We hypothesize that either activation of a deubiquitinase or inhibition of an ubiquitin ligase could potentially explain the reduced ubiquitination and degradation of CAST. Future studies should elucidate this interesting mechanism. The following sentence was added to the results section.

“These results collectively suggested that CAST had a shorter half-life in HD cells than WT cells; CAST was degraded via the UPS, with CAST ubiquitination and degradation both prevented by CHIR99021. CHIR99021 treatment had no effect on proteasome activity *in vitro* and in HdhQ7/111 cells (Fig. S6H, I) nor on MCL1 protein levels, a well-known proteasome substrate⁴⁴ (Fig. 6A, B; Fig. S6F), thereby excluding a global inhibition of CHIR99021 on proteasome activity. Taken together, our results led us to propose a model that CHIR99021 treatment prevents the UPS-mediated degradation of CAST in the context of HD.” (page 13)

We also added the following discussion to address the reviewer’s comment.

“To our knowledge, ours is the first study to reveal the UPS-mediated degradation of CAST. In HdhQ111 cells, the CHX-induced degradation of CAST was prevented by treating cells with selective proteasome inhibitors, suggesting a high turn-over rate for the CAST protein mediated by the proteasome. Although CHIR99021 treatment prevented CAST loss in HdhQ111 cells, it did not improve MCL1 protein levels - a protein degraded by the UPS pathway⁴⁴, consistent with the observation that CHIR99021 did not affect proteasome activity. These lines of evidence suggest that CHIR99021 may selectively modulate CAST degradation rather than “bulk” protein degradation by the proteasome. CAST can be cleaved and degraded by certain proteases, including calpain⁵⁶. However, our results excluded the possibility that CHIR99021 directly affected calpain enzymatic activity. Because the accumulation of CAST ubiquitination

was abolished upon CHIR99021 treatment in HdhQ111 cells, we speculate that CHIR99021 might inhibit ubiquitin ligases or stimulate deubiquitinases that regulate CAST ubiquitination and degradation. Therefore, CHIR99021 may serve as “a CAST-stabilizer” to suppress calpain-mediated cellular injury in HD. Future investigations into the regulation of CAST ubiquitination in HD are warranted to elucidate detailed pathways where CHIR99021 stabilizes CAST.” (page 17)

Minor points:

Label-free proteomics method. The sample amounts mentioned in the method section should be checked carefully. For example, on page 26, ‘peptide digests (320 mg) were loaded onto a column in an 8 μ L injection volume’ is simply not possible. On page 27, for the power analysis carried out, no details have been provided, and the approach used should be referenced.

Response: We apologize for the incorrect description of the method. We corrected the sentence as follows. In addition, we provided detailed power analysis method and related references in Method section (see page 29-30 in the revised manuscript).

“The 300 ng of lysed protein from each of group were loaded onto a column in a 3 μ L injection volume with blanks in between for a total of four LC/MS/MS runs.” (page 29)

“The proteins were identified with Mascot (Matrix Sciences, London, UK). Key search parameters were Trypsin for enzyme; a maximum of 1 missed cleavage; peptide charge states of +2 to +3; peptide tolerance of 10 ppm; and MS/MS tolerance of 0.8 Da. Oxidation of methionine was a variable modification. Identifications were merged, the spectra summed and then quantified using spectral counting (selecting normalized total spectra) in Scaffold version 4.4.0 (Proteome Software, Inc. Portland, Oregon). The protein probability was determined using a protein threshold of 99%, two peptides and a peptide threshold of 95%^{69, 70}. A false discovery rate (FDR) of 0.86% was calculated using ProteinProphet algorithm for the 56,902 spectra examined⁷⁰. Both MASCOT and Scaffold strategies utilized the Uniprot human database from June 2016 (20,199 sequences).” (page 30)

Whereas the abstract and parts of the introduction & results are easy to read, other parts of the manuscript, in particular the methods section, deserve improvements in editing and English language.

Response: We improved the English throughout the manuscript.

Reviewers' Comments:

Reviewer #1:

Remarks to the Author:

The authors have addressed all my comments.

Reviewer #2:

Remarks to the Author:

The authors made substantial revisions to the manuscript and provided additional controls and new data. Most significantly, the data on the CAST-dependent effects of CHIR, data quantification per animal basis, and biochemical evidence of reduced Htt aggregation upon treatment with CHIR further strengthened the manuscript. Overall, the manuscript should be suitable for publication in Nature Communications.

Reviewer #3:

Remarks to the Author:

This reviewer is satisfied with the manuscript revisions provided by the authors and at this point, would recommend publication.

Response letter

RE: Small-molecule suppression of calpastatin degradation reduces neuropathology in models of Huntington's disease (NCOMMS-20-48977A)

Response to reviewers' comments

Reviewer #1 (Remarks to the Author):

The authors have addressed all my comments.

Reviewer #2 (Remarks to the Author):

The authors made substantial revisions to the manuscript and provided additional controls and new data. Most significantly, the data on the CAST-dependent effects of CHIR, data quantification per animal basis, and biochemical evidence of reduced Htt aggregation upon treatment with CHIR further strengthened the manuscript. Overall, the manuscript should be suitable for publication in Nature Communications.

Reviewer #3 (Remarks to the Author):

This reviewer is satisfied with the manuscript revisions provided by the authors and at this point, would recommend publication.

Response: We thank the reviewers for their comments and enthusiasm on our revised manuscript.

Responses to the editorial requests

SUBMISSION INFORMATION

In order to accept your paper, we require the following:

- A revised author checklist describing your response to our editorial requests (attached).

Provided

- A separate point-by-point response to the reviewers' comments, reproduced verbatim.

All three reviewers expressed satisfaction with the revisions made to our manuscript (see above). In this response letter, we thanked the reviewers for their comments and responded to the editorial requests.

- The final version of your manuscript as a Word or LaTeX file, with all changes highlighted in the text and any tables prepared using the table menu in Word or the table environment in LaTeX.

The final version of our manuscript is provided as a Word. All changes are tracked.

- If using LaTeX, please use numerical references only for citations, and include the references within the manuscript file itself. If you wish to use BibTeX, please copy the reference list from the .bbl file, paste it into the main manuscript .tex file, and delete the associated \bibliography and \bibliographystyle commands.

N/A

- The complete author list provided in the manuscript file, which must match that given on our manuscript tracking system. The author list in the main manuscript file will be used during typesetting of your article.
Done

- Production-quality versions of each figure as a separate file containing all panels. To ensure the swift processing of your paper, please provide the highest quality versions of your images and when combining different figure parts into one file for layout, use a vector-based application such as Adobe Illustrator or Microsoft Powerpoint. We recommend .ai, .eps, .pdf, .ppt. Figures divided into panels should be labelled with a lower-case, boldface 'a', 'b', etc. in the top left-hand corner. If resolution is not of sufficient quality, production of your paper will be held whilst replacement files are obtained. For detailed guidance on figure preparation, see <https://www.nature.com/documents/aj-artworkguidelines.pdf>
Provided

- Please note that we do not modify the text in figures to conform to style during the production process. Please ensure that your figures are presented accurately and adhere to the guidance provided.
Done

- Any updated checklists that verify compliance with our research ethics and data reporting standards in PDF format.
Done

- The final version of the Supplementary Information in one PDF file.
Done

- Any Supplementary Movie, Audio, Data and Software submitted as separate files. Supplementary Data and Source Data must be provided as .xls, .xlsx or .zip files, while Supplementary Software must be supplied as .zip files..

** Please note that we do not edit Supplementary Information files; they must be finalised prior to acceptance of the paper. **

We do not have such files in the manuscript.

- If you wish, an interesting image (but not an illustration or schematic) for consideration as a Featured Image on the Nature Communications homepage. The file should be 1200x675 pixels in RGB format and should be uploaded as a Related Manuscript File. In addition to our home page, we may also use this image (with credit) in other journal-specific promotional material.

- Completed and signed copies of our Multimedia License to Publish (LTP) for any Featured Image suggestions (please use one form for each image and give a scientific description of the image in the 'title' field; do not use "Featured Image" as a title): <http://www.nature.com/documents/snl-multimedia-ltp.docx>

There is no image submitted.